# Sense of agency at a temporally-delayed gaze-contingent display

**Junhui Kim**[ORCID]*, **Takako Yoshida**

School of Engineering, Tokyo Institute of Technology, Meguro City, Tokyo, Japan

* kim.j.ba@m.titech.ac.jp

## Abstract

The subjective feeling of being the author of one's actions and the subsequent consequences is referred to as a sense of agency. Such a feeling is crucial for usability in human–computer interactions, where eye movement has been adopted, yet this area has been scarcely investigated. We examined how the temporal action–feedback discrepancy affects the sense of agency concerning eye movement. Participants conducted a visual search for an array of nine Chinese characters within a temporally-delayed gaze-contingent display, blurring the peripheral view. The relative delay between each eye movement and the subsequent window movement varied from 0 to 4,000 ms. In the control condition, the window played a recorded gaze behavior. The mean authorship rating and the proportion of "self" responses in the categorical authorship report ("self," "delayed self," and "other") gradually decreased as the temporal discrepancy increased, with "other" being rarely reported, except in the control condition. These results generally mirror those of prior studies on hand actions, suggesting that sense of agency extends beyond the effector body parts to other modalities, and two different types of sense of agency that have different temporal characteristics are simultaneously operating. The mode of fixation duration shifted as the delay increased under 200–ms delays and was divided into two modes at 200–500 ms delays. The frequency of 0–1.5° saccades exhibited an increasing trend as the delay increased. These results demonstrate the influence of perceived action–effect discrepancy on action refinement and task strategy.

## Introduction

When tackling a task during daily life, one naturally experiences the sensation of "I" am doing a task, which is a subjective feeling of being the author of one's own actions and the cause of the ensuing consequences; indeed, this is referred to as the sense of agency [1]. Those subjective feelings contain mechanisms that properly distinguish "I" from "other" when judging the authorship of an event [2, 3]. A well-known phenomenon related to this feeling is the cancellation of the self-tickling sensation [4–7]. Furthermore, as the temporal gap between action and sensation increases, this phenomenon decreases [8], suggesting that the attribution of a perceived event can be manipulated.

**Data Availability Statement:** Availability of data and materials The data for these experiments are available via the Open Science Framework at https://osf.io/x5w3d/?view_only=13a59dd61add49dfa0f2dabc90f75903 Code

availability: The code for the experiments is available via Open Science Framework at: https://osf.io/x5w3d/?view_only=13a59dd61add49dfa0f2dabc90f75903.

**Funding:** Takako Yoshida was supported by a ROBOT Industrial Basic Technology Collaborative Innovation Partnership grant from the New Energy and Industrial Technology Development Organization of Japan (Grant number JPNP20016). The funders had no role in study design, data collection and analysis, decision to publish, or preparation of the manuscript. https://www.nedo.go.jp/english/activities/activities_ZZJP_100188.html Junhui Kim was supported by the Japan Science and Technology Agency's Support for Pioneering Initiated by the Next Generation (SPRING) Program (Grant Number JPMJSP2106). The funders had no role in study design, data collection and analysis, decision to publish, or preparation of the manuscript. https://www.jst.go.jp/jisedai/en/ There was no additional external funding received for this study.

**Competing interests:** The authors have declared that no competing interests exist.

The "comparator model" has been proposed as an internal model to account for the sense of agency [9–12]; it is grounded in the concepts of efference and afference signal [13]. According to this model, when we choose to execute an action with a specific intent, a signal is dispatched to the muscles, while its replica (efference copy) is retained in order to make a prediction. The external event following our action is then perceived by comparing the event itself (afferent signal) with the prediction we made. Within the comparator model, we acknowledge ourselves as the initiator of an event when the afferent signal aligns with an efferent copy, thereby canceling out the self-originated sensation. This central cancellation has been applied to explain sensory attenuation phenomena, including self-tickle cancellation [4–7], and in visual domain studies. Examples in the visual domain include decreased sensitivity to self-generated Gabor patches [14], diminished event-related potentials in response to self-generated flashes [15], and the maintenance of visual stability despite rapid eye movements [16, 17]. Conversely, any inconsistency between the two signals suggests the potential role of another individual or agent in initiating an event. This comparison between afferent and efferent signals not only serves to cancel the self-triggered sensation but also helps determine the originator of the event, while any inconsistency is employed to refine future motor instructions to achieve the desired outcome [18]. Deficits in such error correction have been reported as a symptom of patients with schizophrenia [9]. In this model, the alignment of efferent and afferent signals plays a crucial role in shaping our perception of self-attribution in external events.

The comparator view has been supported by empirical studies such as those focused on intentional binding [19] and active rubber hand illusion [20], as well as studies that have systematically manipulated a temporal delay between a participant's action and the subsequent event [21–27]. Intentional binding, characterized by the perceived shortening of time between a voluntary action and its subsequent event, was first documented by Haggard et al. [19] and has since been extensively explored, particularly in the context of research on the sense of agency [28]. In the active rubber hand illusion experiment, a rubber hand placed on a box was visible to participants, while their own hand was concealed inside the box and covered up to the shoulder. The participant's index finger was connected to the rubber hand's finger via a rod. Participants reported a reduced sense of agency when the experimenter controlled the rod, thereby moving both fingers, compared to when they controlled it themselves. Furthermore, the sense of agency also diminished when the rubber hand, now detached and controlled by the experimenter, mimicked the participants' finger movements in a temporally incongruent manner, unlike when those participants were in control. Further, the reported degree of agreement regarding the sense of agency tended to decrease, as the participant's action and the monitor's visual feedback were set to have a temporal discrepancy [21–27].

There exists extensive research on the sense of agency, yet few studies focus on eye movements as a means to control computer interfaces. Specifically, the role of action-outcome temporal discrepancy as an independent variable has not been thoroughly explored. Several studies have examined scenarios in which an individual experiences a sense of agency through social gazing, such as the perception that one's own gaze caused another agent's gaze to follow, and have explored how temporal delays in the other agent's gaze response affect the sense of agency [29–31], finding that the degree of agreement in the sense of agency diminishes as the delay increases. However, the use of temporal delays in feedback from a gaze-contingent paradigm specifically for controlling computer interfaces has been largely understudied. Grynszpan et al. [32] and Grgič et al. [33] demonstrated that a sense of agency can be experienced through eye movements using a gaze-contingent experimental design. Nonetheless, their investigation into temporal delays as an independent variable was limited.

Our study aims to bridge the aforementioned gap by exploring how delayed feedback in eye-gaze human-computer interaction (HCI) affects users' subjective authorship and behavior.

We hypothesize that the effect of action-feedback temporal delay on the sense of agency in eye-gaze HCI will resemble that observed in limb modalities. First, participants will report themselves but not the other agent (i.e., computer) as the event author when there are temporal delays in eye-gaze HCI, as seen in previous findings [22]. Second, if we measure the intensity of their sense of authoring, it will decrease in response to those temporal delays around up to around 300–500 msec, as was seen in the results with limb modalities [21, 24–27]. The above hypothesis is grounded in the understanding that the predictive process based on the efference copy involves not only foot and limb actions [4, 5, 8, 34, 35] but also eye movements (i.e., corollary discharge) in the context of perceiving a stable world despite rapid gaze shifts and consequent retinal image shifts [13, 16, 17, 36]; indeed, this was illustrated in a study with monkeys, where the inactivation of the corollary discharge circuit altered visual stability [37]. Similarly, Lindner et al. [38] demonstrated, in their study, that the ability to discriminate one's own smooth eye pursuit from motion perception was worse in schizophrenia patients. We are also interested in whether different modalities, eye movements versus limb movements, lead to distinct dynamics in the reported sense of agency or not. While the eyes were not originally intended for direct manipulation of computer interfaces, this unique characteristic of the eye-gaze modality might not yield different responses to temporal delays. Should our findings align with those of previous studies, they will contribute to broadening existing models of the sense of agency, as a relatively modality-independent action-feedback comparison process.

Our interest in the impact of temporal delays is significant not only from a theoretical perspective but also from a practical standpoint. Sense of agency has been addressed as a crucial aspect in evaluating a user's experience in HCI [39], with a focus on the user's feeling of control over the system. Given the emergence of eye-gaze as a new HCI modality [40], and the growing availability of accessible eye trackers, e.g., Tobii [41] and Windows 10's eye movement control features [42], understanding the above-mentioned impact becomes increasingly vital. Delays in gaze measurement and rendering can reach up to 500 ms in remote applications such as remote surgery [43], and several seconds in teleoperation over complex Internet connections [44, 45]. Moreover, direct communication with orbiting space stations, such as the International Space Station, can encounter a round-trip time of up to 1.2 seconds [46], and up to 7 seconds when operating robotic arms on satellites [47]. In the current experiment, we set temporal delays in the eye-gaze HCI's update at 0 to 4000 ms, as such extended delays, simulating teleoperation scenarios, have not been tested in the sense of agency studies.

In designing our study to measure the sense of agency, we did not adopt the intentional binding paradigm. It involves two paradigms: single-event time estimation and interval estimation. The Libet clock method, a variant of single event time estimation, requires participants to report the clock hand's position when they press a button or hear a corresponding audial cue [19]. The other method involves reporting the estimated time interval between action and feedback [24, 26]. Haggard et al. [19] found that, with the Libet clock, the time interval was perceived as shorter when participants voluntarily pressed the button, in contrast with when the action was involuntarily induced by transcranial magnetic stimulation, thus highlighting the sense of agency. Now intentional binding is regarded as one of the behavioral methods to assess the sense of agency without questionnaires [28].

Given that one of our goals was to generalize our findings to teleoperating HCI contexts, we adopted a visual task involving continuous eye movements, instead of adopting the intentional binding paradigm with a single event. We were particularly cautious of the complexities introduced by the chronostasis effect, which can complicate an experimental design. This effect, detailed by Yarrow et al. [48], is the dilation of perceived time immediately following a saccade. It becomes problematic in a task simplified to a gaze-induced button click, whereby a single-button click task is a typical format in intentional binding studies [28]. Gutzeit et al.

[49] investigated the diminishing effect of intentional binding with increasing temporal delays within a gaze-contingent paradigm, yet further research is necessary with a higher degree of freedom in eye movements (i.e., continuous eye movements rather than a single saccade). Also, the research gap highlights the necessity to explore how users alter their continuous eye movements with behavioral strategies as their sense of authorship diminishes. Based on these previous findings and discussion, instead, we opted for a questionnaire approach. While questionnaires have been criticized for only capturing the partial aspect in the sense of agency rather than its full extent [50], they still provide sufficient information to assess our hypothesis. The aforementioned approach also serves as a reasonable starting point for exploring this previously uninvestigated topic, allowing us to compare our results with those of previous studies.

Instead of using the simple two-alternative forced choice asking "self" or "other" as the agent of the observed action for the participants, we adapted Farrer et al.'s [22, 23, 51] method of measuring agency by offering a choice between three responses: 'eye-gaze HCI's updates corresponded to my eye movements' (self), 'corresponded to my eye movements but with delay' (delay), and 'eye-gaze HCI was manipulated by computer system' (other). In our study, the 'partial control' option from Farrer et al. [22, 23, 51] was modified to 'delay,' specifically to reflect the controllable object of our HCI. Farrer et al. [22, 23, 51] have proposed that the two choices of "self" and "other" do not fully cover scenarios where participants perceive their actions as contributing to an outcome, but not completely or directly (i.e., "I perceived that I controlled the outcome, but the causal relationship between my actions and the outcome was temporally distorted"). As such, we introduced the "delay" option, in case participants experienced a sense of agency influenced by, but not entirely negated by, temporal delays in the interface's response. In addition to temporally delayed conditions, we established an uncontrollable condition, as recommended by Farrer et al. [51], to assess whether this questionnaire could capture participants attributing to another agent. In adapting Farrer et al.'s [22, 23, 51] method of measuring agency, we aimed to replicate their established approach, expecting our questionnaire to yield similar results in the context of eye-gaze HCI.

Farrer et al. [22] found that participants did not attribute events to another agent even with a 1,000 ms delay between their action and visual feedback when they adopted three-choice responses instead of two. Synofzik et al. [50] proposed a revised model, incorporating but expanding beyond the comparator model, which they described as incomplete to fully explain the sense of agency. Synofzik et al. [50] proposed a two-component model of the sense of agency: a non-conceptual feeling of agency, distinguishing the author of the action, and an explicit conceptual judgment of agency, interpreting the author of the event based on contextual or conceptual cues, such as intentions and beliefs. In Farrer et al. [22], the participants were instructed that the experimenter might control the stimuli and that they, the participants, could not. In the current study, we instructed participants that when they could not control the stimuli, the computer system might be in control. By almost perfectly adapting their instructions and thereby providing similar contextual cues of computer behavior, we predicted that our investigation of the delay's effect on the sense of agency would well resemble the results of Farrer et al. [22], partially because of the participant's belief that the computer system can control the visual display.

In eye-gaze HCI, when discussing the importance of the sense of agency, it is crucial to clarify the type of agency involved. We believe that the type of agency related to the classical comparator model in visual stability research should be distinct from that involved in situations like waiting for a coffee machine or gaze typing and button press in eye-gaze application originally for amyotrophic lateral sclerosis (ALS) patients, where a minute-long delay can be acceptable. The essence of agency in gaze-contingent systems lies in the immediate and continuous nature of the interaction, where any significant delay disrupts the user's experience. A

delay of even a few seconds in such systems can entirely break the first-person, immersive experience that is central to effective eye-gaze interaction. While some might argue that a belief-based sense of agency is sufficient for assessing the quality of HCI, prolonged delays shift the interaction model from natural, real-time control to more premeditated or anticipatory actions, which undermine the intuitive and direct control expected in these systems. As has been observed in first-person shooter (FPS) video games, minimal delays are crucial to preserving a seamless and immersive experience [52]. Similarly, in gaze-based interfaces, even minor disruptions in temporal proximity can force users out of a natural interaction flow, making the system feel less responsive and significantly degrading user satisfaction.

In addition to the categorical questionnaire described above, we further adapted a rating questionnaire, with our focus being on assessing the influenced sense of agency within the spectrum of 'I' as the event author, excluding the "other" option. The question was a six-point rating scale ranging from 6, full manipulation ('I fully manipulated the window') to 1, a lack of control ('I could not possibly manipulate the window'). While our questionnaire resembles those employed in prior studies [21, 24–26], it offers clarity by avoiding 'disagree' responses, which could be misconstrued as implying control by another agent. Instead, our questionnaire directs participants to exclude the "other" consideration when rating the degree of authoring. By not including the "other" category from the scale, we intended to more reliably investigate the variations in the perception of "self" as the author. While the "delay" response in categorical questionnaires serves the role of indicating altered self-authoring feelings, there is a high possibility that "delay" responses will converge to a high proportion at low temporal delays (prior research suggests around 300 ms [22]) and remain stable throughout longer delays. This can potentially oversimplify participants' perceptions in extended delay conditions. Continuous scales provide granularity, measuring how the perceived authorship of 'I' decreases throughout our delay conditions over 300 ms. With these two questionnaires, we posit that we can capture both the clear distinctions in altered 'I' authorship and the detailed variations in attenuated 'I' authorship within the context of eye-gaze HCI.

We assume that participants will adapt their strategies in response to these delays. This assumption aligns with the comparator model, which posits that discrepancies between predicted and actual outcomes are used to adjust motor commands in order to complete tasks [18]. In this study, participants conducted a visual search with our eye-gaze HCI that forced them to search in a serial manner, aiming to effectively disrupt their behavior with delays. With our HCI, a character is readable only within a controllable window that follows the user's eye position, blurring the remaining outside area. Such disruption in the practical applications of this HCI is a critical aspect to consider. Through our measurements with two questionnaires, we sought to identify a critical temporal delay—a threshold at which a user's experience becomes notably disrupted in eye-gaze HCI. Previous studies have shown that participants notice visual artifacts in eye-gaze HCI at delays of 50–70 ms [53, 54]. The threshold for disruption in users' sense of authoring may vary.

In eye-gaze HCI research, gaze-contingent multiresolutional rendering [55] is used as "foveated imaging" or "foveated rendering" [53, 56–59]. This technique produces frames with multiple-resolution regions and blended edges, considering the visual acuity of the human eye from the fovea to the retina's periphery [60]. Foveated rendering reduces data processing while preserving the user's visual experience [61] and is commonly used in virtual reality headsets with high display resolution demands [57–59, 62]. In this experiment, we employed a simple bi-resolution gaze-contingent display composed of a high-resolution and blurred display, featuring an unblended border contrary to other studies [61, 63–65]. This design emphasizes the variance in saliency with our HCI and aims to provide valuable insights for optimizing interface design and enhancing user experience in a variety of HCI environments. In addition

to exploring the effects of temporal delays, we introduced variations in the saliency (i.e., visibility) of the window. Initially, a high contrast was implemented between the window and the surrounding area. Subsequently, we adjusted the design to more closely align the visibility between these two elements. This methodological decision, involving two distinct levels of visibility, was intended to facilitate the generalization of our findings.

Building on previous work that examined the sense of agency in gaze-contingent displays with inconsistent, jittery delays designed to mimic variable packet transfer over the internet [66], this study specifically investigates the effects of controlled, consistent temporal delays on participants' sense of agency and behaviors. The experimental setup, including the dynamic gaze-contingent window, was adapted from the earlier research [66].

Overall, the study was driven by the following three objectives:

1. Evaluate the impact of temporal delays on participants' perceptions of event authorship, categorizing their responses as self, delayed self (delay), or other.

2. Examine how temporal delays influence the intensity of subjective feelings of authorship, as assessed through a rating scale.

3. Investigate the effects of temporal delays on participants' behaviors.

As a side objective of 1 and 2, we sought to identify the critical threshold of temporal delay at which participants' experiences begin to alter.

## Method

### Participants

Because of the Chinese characters in the stimuli, only the individuals who could read Chinese characters were recruited. In total, 11 participants were recruited for the study, with three participants being excluded during the experiment since the calibration accuracy was over 1˚. Data from eight participants (average age: 24.4 years; standard deviation [SD]: 1.5; range: 22–27 years; one woman and seven men: five Japanese, two Koreans, and one Chinese) were analyzed. All the participants were students (two undergraduate students and six master's program students) from the Tokyo Institute of Technology. Undergraduate students were informed of the experiment in their classrooms, while master's program students were informed via their advisors through e-mails from the experimenters. The participants received a 1,000-yen book voucher card for every hour of participation in the experiment. The participants had normal or corrected-to-normal vision. Written informed consent was obtained from all the respondents before the start of the study, and the work was approved by the Tokyo Institute of Technology Human Subjects Research Ethics Review Committee. We collected data from October 6, 2020, to January 26, 2021, after which time we conducted statistical analyses and assessments to address our research questions. Following the guidelines provided by Lakens [67], we acknowledge the absence of a formal sample size justification. Our sample size was limited due to resource constraints. Participants were restricted to one hour participation (approximately 6 sessions) per day by ethics guidelines and preferred to participate only one day per week. Thus, each participant needed about ten days over two months to complete the 60 sessions. When potential participants learned about the required times and examples from others, many hesitated to join, making recruitment challenging.

### Apparatus

Experiments were conducted in a dimly lit room to allow participants to re-position their hands during their responses. Participants were asked to stabilize their heads using a chin rest

during the experiment. Visual stimuli were displayed on a 24" LCD monitor (BenQ Corp., XL2411t) at a resolution of 1920 × 1080, a frame rate of 144 Hz, and a viewing distance of 60 cm. The monitor's input lag is theoretically 7 ms, given the 144 Hz display. Eye movements were monocularly recorded (left eye) using an EyeLink 1000 Plus eye tracker (SR Research Ltd.), with a spatial resolution of <0.02˚ and a sampling rate of 2,000 Hz. The eye tracker's sample delay was less than 1.4 ms on average (SD <0.4 ms). The minimum temporal delay from an eye movement to the monitor's feedback includes the eye tracker's sample delay, the processing time of the experiment software (updated every 1 ms), and the display's input lag. Participants used a numeric keypad on a keyboard (Lenovo Group Ltd., SK-8825) to deliver their responses. The experiment software, modified from the GCWINDOW template by SR Research Ltd., was designed for collecting eye movement data, buffering, and updating stimuli with a temporal delay. For one specific trial per session, the software stored eye movement data to be replayed later. While frames were rendered every 7 ms, data acquisition and internal computations were performed every 1 ms. Nine-point calibration and validation were conducted, with a validated accuracy of <1˚.

## Stimuli

In our eye-gaze HCI, we implemented peripheral blurring during participants' visual search, where Chinese characters were readable only within a controllable window. We aimed to require participants to search sequentially. Additionally, this design aimed to disrupt their behavior with delays effectively. However, our HCI has an inherent concern that saccade amplitudes have been shown to decrease with peripheral blurring [61, 63–65, 68]. For instance, Reingold and Loschky [61] observed delayed first saccades and reduced saccade amplitudes when participants used a peripheral blur window while tracking a moving object in a video. It has been proposed that such effects on eye movements are influenced by reduced saliency in peripheral targets [69]. These findings are pertinent to our study for two reasons: firstly, dominant eye-gaze HCI systems employ peripheral blurring [53, 56–59], and, secondly, we considered the window's effect as being beyond our study's scope.

To counter the unintended effect, decreasing saccade amplitudes, we created a distinct stimulus. Despite the expected decrease in visibility within the blurred periphery of our stimuli, we anticipated a lesser degree of reduction compared to natural stimuli, thereby helping to mitigate the influence of peripheral blurring. A visual search array (Fig 1A) with nine black (luminance: 4.71 ± 1.32 $cd/m^2$) Chinese characters were presented in a 3 × 3 virtual matrix on a gray background (luminance: 30.41 ± 3.15 $cd/m^2$). Each character had a size of 3˚. The horizontal and vertical distances between two nearby characters were 11˚.

The low visibility condition was achieved by blurring the periphery, whereas the high visibility condition involved both darkening and blurring the periphery. There were two types of blurred images: one in which only blurring was applied (Fig 1B) and one in which both the darkening and blurring processes were applied (Fig 1C). For the image blurring, a Gaussian filter was applied to one image randomly selected from the 540 images (Fig 1B and 1C) using Adobe Photoshop 2020. The low-pass filter was applied with a 10-pixel (.2˚) standard deviation, while the signal was attenuated by 3 dB at a spatial frequency of 0.57 cycles/˚ and was further attenuated at higher spatial frequencies. The blurred Chinese characters had increased luminance (15.27 ± 3.76 $cd/m^2$), while the luminance of the gray background was maintained. The darkening process was performed by decreasing the red-green-blue (RGB) brightness (0–255, 0–255, and 0–255) of the blurred image by 54 for each value. As a result of reduced brightness, the brightness of the blurred image decreased in RGB brightness from 231 to 177 (background luminance: 20.28 ± 4.29 $cd/m^2$, character luminance: 5.21 ± 2.04 $cd/m^2$).

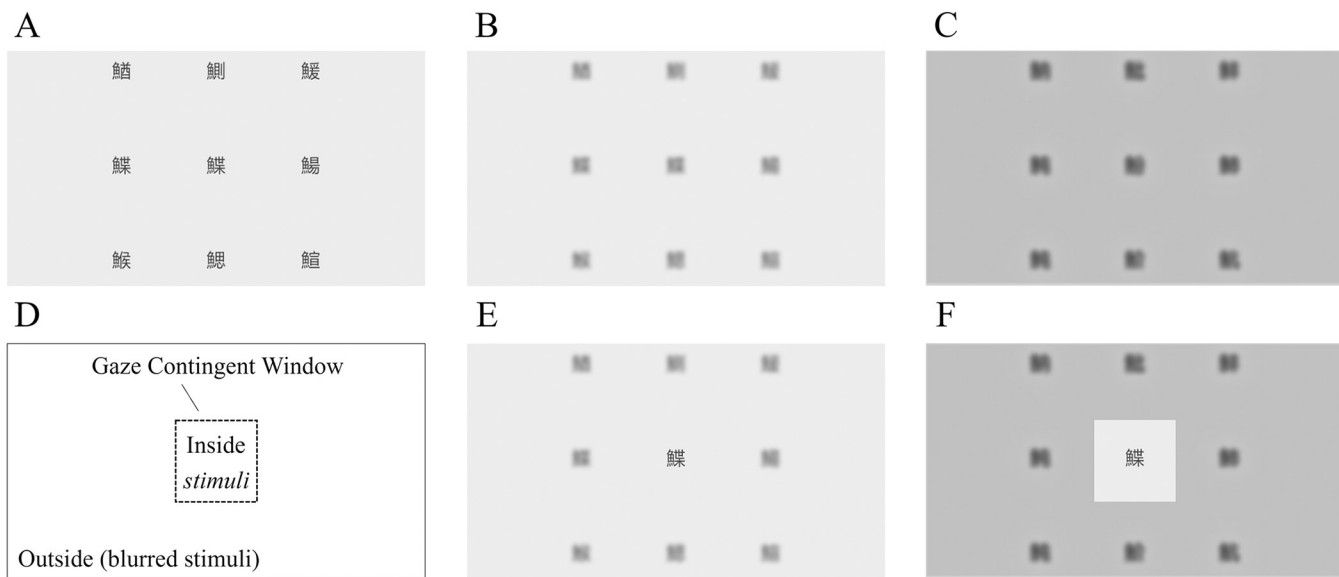

**Fig 1.** A) An example of the image produced by random image generation; 540 images were generated. B) One of the images was randomly selected and blurred. C) In addition to the blurring, darkening was applied. D) A virtual square window was set at the center of the participant's fixation location. A clear image was displayed within the inner side of the window, while the outer side of the window presented a blurred image. E) Example of the stimulus with the blurred image (Fig 1B) when the participant fixated their gaze on the central Chinese character (low visibility condition). F) Example of the stimulus with the blurred and darkened image (Fig 1C) when the participant fixated their gaze on the central Chinese character (high visibility condition).

A 10˚ square window was dynamically centered at the participants' fixation location in accordance with their eye movements (Fig 1D). The edge of the window was 5˚ vertically or horizontally from the fixation location. The Chinese character array (Fig 1A) was rendered on the inner side of the gaze-contingent window, whereas one of the blurred images (Fig 1B and 1C) was rendered on the surrounding outside portion (Fig 1D), illustrating a distinct border between two regions with a step function. An example of a presented image with a blurred exterior is shown in Fig 1E (low visibility), while an example of a presented image with a blurred and darkened outside is shown in Fig 1F (high visibility). Although we opted to create a display in which only one character could be read with fixation, the participants could strategically locate their gaze between the two characters to view both simultaneously. Our analysis revealed that the participants did not execute such a strategy (see S4 Appendix in S1 File).

Chinese characters were pooled based on two rules (rather than using the English alphabet; see S3 Appendix in S1 File for a pilot study comparing English and Chinese characters). First, each Chinese radical character had to be part of a fish name to ensure that all characters' meanings were within the same fish category and to thus prevent meaning- or category-based attentional bias. Second, the stroke counts of the nine characters had to be the same to ensure that they shared similar spatial frequency, contrast, complexity, and visibility in human peripheral vision. A total of 239 Chinese characters (see S1 Appendix in S1 File) were selected for this study.

For the visual search array (Fig 1A), we set a 50% probability for the presence of a target character among the surrounding eight locations, resulting in two types of arrays (target and non-target). The uncertainty was introduced to discourage participants from predicting the target's location before completing the visual search and potentially affecting the participant's behavior. The correct response rate for individual tasks was 96% at its lowest, indicating that participants engaged with each trial without resorting to prediction (see S2 Appendix in S1 File for response rate per participant and condition). Using preprogrammed random image-

generating Unity software, 540 image files were generated. However, because we did not sample the images, the probability of the target's existence and location was uneven. Of the 540 images, 272 were non-target arrays. With regard to the 268 target arrays, the target counts at each of the eight locations were as follows (from the middle top, described clockwise): 34, 29, 32, 32, 35, 40, 37, and 29.

## Task and procedure

Participants were instructed to locate a target character, matching the one at the center, within the eight peripheral locations. Participants had to press the corresponding button at the numeric keypad when the target character was present or '0' when the target character was absent. Participants were instructed that the gaze-contingent window has a delay following their eye movements and is occasionally controlled by the computer system. We asked participants to conduct the visual search even when they could not manipulate the window, and to press '0' if they could not find the target in such an uncontrollable trial. The task terminated with the participants' response via the Enter key.

Following each task, participants had to answer the two questionnaires. They had to report whether they perceived that they controlled the movements of the window with their gaze (self), perceived that the causal relationship had a temporal distortion, which is a delay (delay), or perceived that the movements were unrelated to their gaze but controlled by the computer (other) (this was the categorical authorship questionnaire). Additionally, participants were asked to rate the extent to which they felt that their eye movements caused the window to move. They reported on a six-point rating scale, where six meant, "I fully manipulated the window," and one meant "I could not possibly manipulate the window" (this was the authorship rating questionnaire).

As shown in Fig 2, a trial was started with a drift check. The visual search began after displaying the blurred image for 4,680 ms. The duration was set to ensure that the experimental software, in the delayed trials, had sufficient time to buffer a participant's gaze behavior. 4,680 ms was uniform throughout the conditions and participants were instructed to fixate on the center character during this time. Then, they had to report two authorship questionnaires.

Sixty sessions (1,080 trials) were conducted for each participant, with 30 sessions (540 trials) conducted for each visibility condition. Participants first completed 30 sessions in one visibility condition before switching to the other. The order of the visibility conditions was

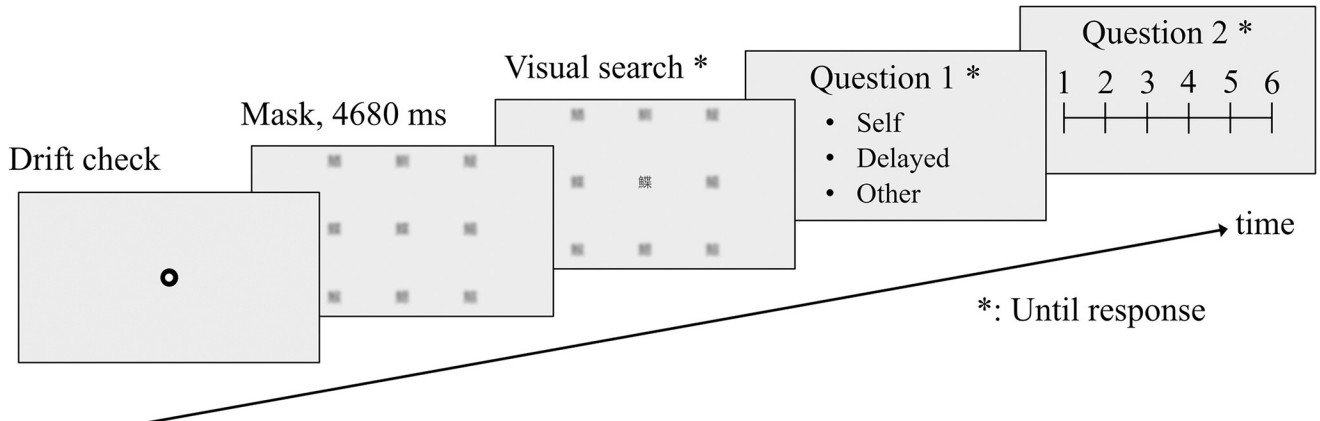

**Fig 2. The sequence of the trial.** After the drift check, participants had to wait until the gaze-contingent window appeared. After a participant responded, they had to answer two questionnaires. Drift checks and questionnaires were exaggerated (size was larger) for explanations.

counterbalanced among participants. A session contained 18 trials with two practice trials, 15 delayed trials, and one playback trial. Fifteen delayed trials and one playback trial were randomly sequenced after two practice trials. The practice trial was identical to a 0 ms delay trial, meaning that it did not introduce any intentional temporal discrepancy. In each delay trial, the gaze-contingent window had one of the following 15 relative temporal delays: 0, 30, 60, 100, 200, 300, 400, 500, 1,000, 1,500, 2,000, 2,500, 3,000, 3,500, and 4,000 ms. The playback trial was set as an uncontrollable condition. The window replayed the participant's recorded gaze behavior in one of the two practice trials, not corresponding to the participant's real-time gaze. In between the visibility conditions, participants performed two sessions (36 trials) without a gaze-contingent display. This 'no window' condition was set to investigate the influence of the window on the participant's behaviors. Under the aforementioned condition, the participants did not answer any subjective questionnaires. Each participant was presented with 36 randomly selected images from a set of 540 images in each session. Participants rested, recalibrated, and were revalidated after each session.

In each visibility session, the same set of 540 images was presented twice, in an identical sequence, to facilitate data preprocessing. Throughout the experiment, we periodically inquired as to whether subjects remembered any stimuli. Given that none of the participants reported any recollection, and considering the sequence counterbalanced between participants, we deemed that the reuse of the stimuli with the same sequence across the visibility conditions did not influence the results.

## Analyses

For all analyses, the alpha level was set at 0.05 to determine statistical significance.

To investigate our first objective, whether the participants correctly inferred the action agent, we analyzed the categorical authorship responses. We employed two generalized linear mixed models (GLMM; SPSS 24.0), excluding playback trials, comparing "self" to "delay" responses and "other" to "delay" responses, using a multinomial probability distribution and a generalized logit link function. For the random effects, we adopted participants with variance components as the covariance structure. Window visibility and temporal delay were analyzed as fixed effects. A total of 42 responses were excluded due to data loss in the subjective report, leaving 7,158 responses for analysis. The Satterthwaite Formula was employed to approximate the degrees of freedom because the sample size was relatively small and the data were unbalanced due to partial data loss.

To investigate the decreasing intensity of the sense of authoring with the delay, which is our second objective, we analyzed the authorship rating. A visibility × delay repeated-measures ANOVA with Greenhouse-Geisser correction was applied to the participants' mean authorship ratings for the delayed window condition. To examine the decreased rating in the playback condition, we combined the two visibility data groups into one since neither visibility nor interaction had a significant main effect (refer to the Results section). We then conducted a paired t-test to assess participants' mean rating scores between the playback and the 4,000 ms delay conditions (each group comprised 16 data samples).

For our third objective, examining the impact on participants' behavior, we first analyzed the response time to investigate whether our eye-gaze HCI forced participants to conduct a visual search in a serial manner, effectively disrupting their behavior with delays. With our target and no-target stimuli, the response time should be less with target stimuli. The time from stimulus onset to the registration of the participant's last input was measured. Inputs other than the last were excluded as the response was often re-registered when participants had to relocate their hands to the keypad. We applied visibility × target × delay repeated-measures

ANOVA with Greenhouse-Geisser correction to the participants' mean response times under the delayed window condition.

Additionally, we examined the effect of the gaze-contingent window. We combined the high/low-visibility groups into one since visibility did not exhibit a main effect (refer to the Results section), and conducted a target × window presence repeated-measures ANOVA with Greenhouse-Geisser correction. The following four data groups were used: 1) no window with no target stimuli, 2) no window with target stimuli, 3) combined data from high/low-visibility windows with 0 ms delay and no target stimuli, and 4) combined data from high/low-visibility windows with 0 ms delay and target stimuli. Each window-presence group contained 16 data samples, whereas each window-absence group contained eight. In this ANOVA, since we have already discussed the effect of the target, we report only the effect of window presence.

Further, we examined the impact of the delay on eye movements, specifically the relative distribution of fixation duration and saccade amplitude. Using SR Research Data Viewer software, saccades, and fixations were defined from raw gaze data. Minimum thresholds for eye movements (0.1˚, 30.0˚/s, and 8000.0˚/s$^2$), which are predefined values in the software, were set for detecting saccades. Fixations outside the stimuli, those that commenced immediately before and after a blink, and those shorter than 120 ms were excluded from the analysis. The mean relative fixation duration frequencies were calculated using the representative frequencies of the eight participants. For each participant and each condition, the frequency of fixation durations within each 25 ms bin in the 0–800 ms range was converted to the relative frequency. The frequency was converted to a relative frequency because the number of eye movements varied significantly among the participants due to individual differences in response time. Additionally, the means of the relative saccade amplitude frequencies were calculated using the representative frequencies of the eight participants. For each participant, the frequency of saccade amplitudes within each 0.5˚ bin in the 0–15˚ range was converted to the relative frequency for each condition.

We conducted additional analyses to examine the changes in the proportion of "self" responses in the categorical questionnaire and behavioral changes in eye movements within the 0–500 ms range (hereafter referred to as contrast analysis) as we observed changes in the range (refer to the Results section). The rationale in selecting the range was that the proportion of "self" responses fell below 25% in the 200–300 ms range and converged thereafter. The rating score in the contrast analysis was not included. The purpose of the contrast analysis was to gain insights by highlighting the significant changes in participants' experiences within the 200–300 ms range. The reported mean rating scale was gradually decreased throughout the delay conditions. The mean rating score put more significance on investigating the variation throughout the delay rather than the 200–300 ms.

Following the observation of a shifting mode with the delay in the fixation duration distribution, we examined the x-axis location of that mode. For each participant and each condition, we calculated the fixation duration distribution using 50 ms bins within the 0–800 ms range. Then, for the bin where the count showed the peak of the shifting mode, we re-located the bin in 1 ms increments, finding the location that showed the maximum count within a range of -50 to 50 ms from the original location. Additionally, it was observed that the saccade amplitude under 1.5 degrees increased with the delay. For the relative distribution of saccade amplitude in each participant and condition, based on the calculated distribution above with 0.5˚ bins, the cumulative distribution for saccades at 1.5 degrees was calculated. Then, the amount of change in each value (proportion of "self" responses, location of shifting mode in fixation duration, and cumulative distribution for saccades at 1.5 degrees) for each 100 ms increase in delay, including 0–100 ms, 100–200 ms, 200–300 ms, 300–400 ms, and 400–500 ms was calculated. Two multivariate ANOVA: one for delay × visibility and another for delay

increase × visibility was conducted. With both ANOVA, post-hoc pairwise comparisons were done with Bonferroni correction. The Compact Letter Display (CLD) graphs were used to plot the significant differences found in the post-hoc analysis.

To examine the critical temporal delay based on the categorical authorship questionnaire, only the proportion of the "self" response at each delay and visibility condition was pooled, then a delay × visibility repeated measures ANOVA with Greenhouse-Geisser correction was conducted. Because the effects of delay and visibility have been discussed above, we only report the post-hoc pairwise comparisons with Bonferroni correction. For the authorship ratings, post-hoc pairwise comparisons with Bonferroni correction following the visibility × delay repeated-measures ANOVA described above were conducted. For the response time, post-hoc pairwise comparisons with Bonferroni correction following the visibility × target × delay repeated-measures ANOVA described above were conducted.

## Results

### Subjective authorship

**Categorical response.** This questionnaire addresses the categorical authorship variable using three levels (self, delay, and others). Fig 3 presents the results.

In the GLMM comparison between the "self" response and "delay" response, there was a significant main effect of delay ($b$ = -5.770, $SE$ = 1.846, $t$ = -3.126, $p$ = .002, Odds Ratio [OR] = .003), as the "self" response was chosen to be less than the "delay" response with increasing delay. Comparing the low visibility window to the high visibility window (LV–HV), the window visibility was significant as a main effect ($b$ = 1.175, $SE$ = 0.378, $t$ = 3.111, $p$ = .002, OR = 3.238). This main effect indicates that the "self" response was chosen the most with a low visibility window. As shown in Fig 3, the participants' "self" response proportion decreased with a relatively longer delay in the low visibility window than in the high visibility window, which implies that the participants detected the delays from a relatively short delay in the high visibility window. This tendency is supported by the significant interaction of Delay × LV–HV ($b$ = -6.223, $SE$ = 1.965, $t$ = -3.168, $p$ = .002, OR = 0.002). All the covariate parameters and fixed coefficients of the GLMM are presented in Table 1.

In the GLMM comparison between the "other" response and the "delay" response, the main effect of the delay was not significant ($b$ = -0.068, $SE$ = 0.213, $t$ = -0.318, $p$ = .750, OR = 0.934), as the "other" response was rarely chosen in the delayed condition. Comparing the low visibility window to the high visibility window (LV–HV), window visibility was not significant as a main effect ($b$ = -0.088, $SE$ = 0.296, $t$ = -0.296, $p$ = .767, OR = 0.916), indicating that the "other" response was barely chosen in both window conditions. An interaction of Delay × LV–HV was not significant ($b$ = -0.080, $SE$ = 0.248, $t$ = -0.322, $p$ = .747, OR = 0.923). All the covariate parameters and fixed coefficients of the GLMM are presented in Table 2.

To examine the critical temporal delay based on the authorship questionnaire, we conducted a delay × visibility repeated measures ANOVA with only the "self" response. Because the effects of delay and visibility have been discussed above, we only report the post-hoc pairwise comparison with the Bonferroni correction. The proportion was significantly different compared to the 0 ms delay (M = .954) when the delay was 200 ms (M = .126, p < 0.001) and above (p < 0.001).

**Continuous response.** The authorship rating was a 6-point rating scale of the degree of authorship. Fig 4 presents the results.

A visibility × delay repeated-measures ANOVA revealed a highly significant main effect for delay (F [2.199, 15.391] = 149.930, p < 0.001, $\eta_p^2$ = .955), while neither visibility (F [1, 7] = .118, p = 0.741, $\eta_p^2$ = .017) nor interaction (F [2.874, 20.120] = .518, p = 0.667, $\eta_p^2$ = .069) was

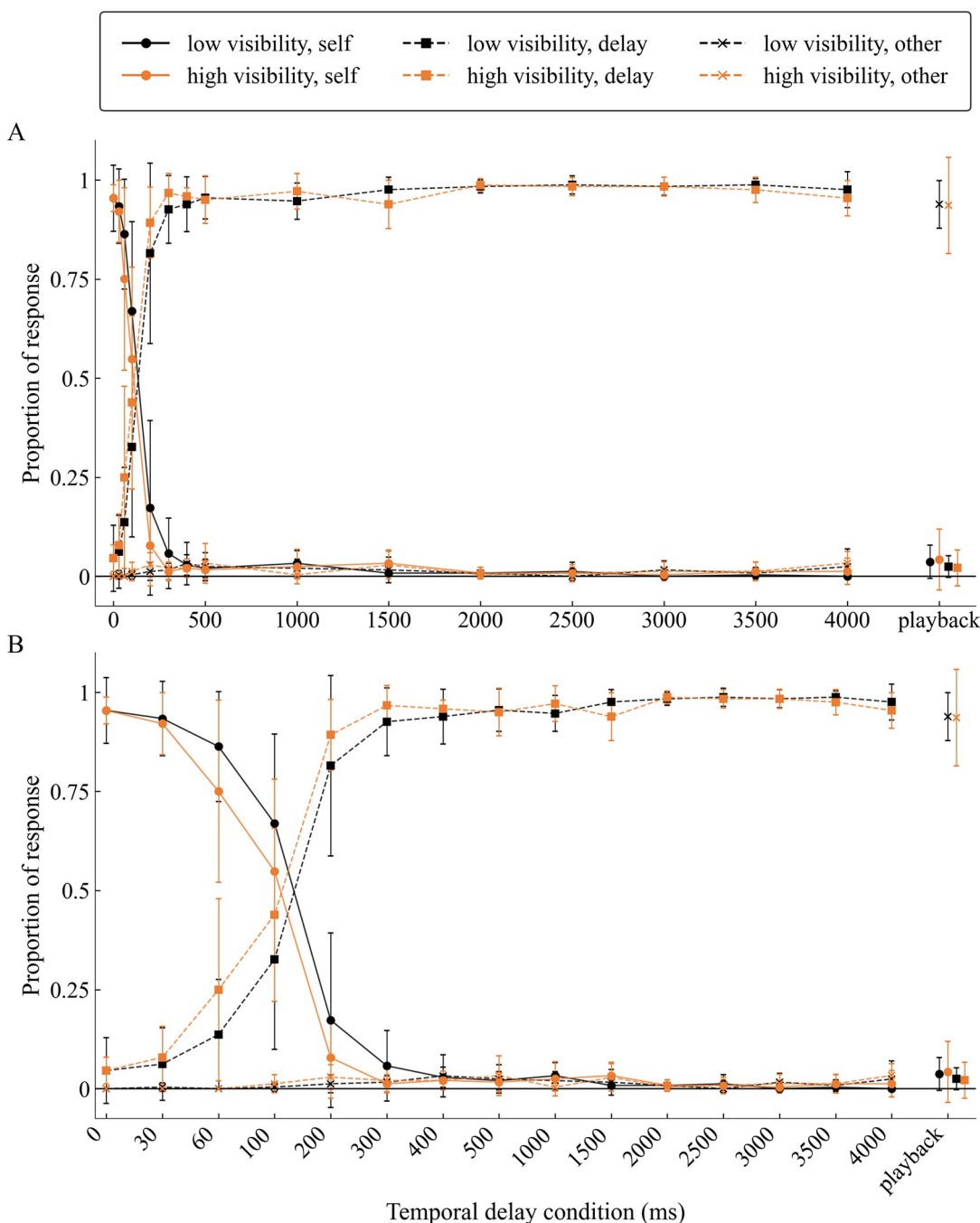

**Fig 3. The proportion of participants who reported authorship (self, delay, or other).** Means and standard deviation were calculated among the participants for each condition and the window visibility. The same results are presented with both a linear x-axis (A) and a nonlinear x-axis (B), which was chosen for comparison with the previous study, in which a similar categorical response was reported with the nonlinear x-axis. [22]. To improve visibility and reduce overlap of points, data points for the "playback" condition have been offset (dodged) horizontally.

significant. With our aim to examine the critical temporal delay, post-hoc pairwise comparison with Bonferroni correction showed that the mean authorship was significantly different compared to the 0 ms delay (M = 5.883) when the delay was 200 ms (M = 4.534, p = 0.031) and above (p < 0.05), indicating a critical temporal delay at the authorship rating.

**Table 1. Estimated covariance parameter and fixed coefficients for the authorship questionnaire, comparing the "self" to the "delay" response.**

| Covariance parameter | Estimate | Standard error | Z score | P value |
|---|---|---|---|---|
| Participants | 0.181 | 0.104 | 1.744 | .081 |
| Fixed effects | Coefficient | Standard Error | T score | P value |
| Intercept | 0.758 | 0.264 | 2.870 | .006 |
| Delay | -5.770 | 1.846 | -3.126 | .002 |
| Low visibility vs. high visibility (LV–HV) | 1.175 | 0.378 | 3.111 | .002 |
| Delay × LV–HV | -6.223 | 1.965 | -3.168 | .002 |

To examine the decreased rating in the playback condition, we conducted a paired t-test to assess participants' mean response times: playback versus 4,000 ms delay. The mean authorship rating was higher in the 4,000 ms delayed window (M = 1.87) than in the playback condition (M = 1.24). The mean difference (0.63) was significant (t [15] = 4.722, p < 0.001, Cohen's d = 1.465), confirming that the mean authorship rating was the lowest in the playback condition.

## Behaviors

**Response time.** The response time was assessed to reflect the altered behavior through the delays and windows. The mean and standard deviations of the response time were computed for the participants' mean response time at each condition, presence or absence of the target, and window visibility (Fig 5).

With the visibility × target × delay repeated-measures ANOVA, there was a significant main effect for delay (F [1.350, 9.450] = 53.906, p < 0.001, $\eta_p^2$ = .885), thus confirming that the response time increased with delay. The main effect of target existence was also significant (F [1, 7] = 18.180, p = 0.004, $\eta_p^2$ = .722), with longer response times for non-target stimuli (M = 9586 ms) and shorter response times for target stimuli (M = 7693 ms). The mean difference (1894 ms) was significant in the post-hoc pairwise comparison with Bonferroni correction (p = 0.004). Visibility was not significant (F [1, 7] = .007, p = 0.936, $\eta_p^2$ = .001). The Delay × target Interaction was significant (F [3.532, 24.723] = 4.265, p = 0.011, $\eta_p^2$ = .379).

We examined the effect of the gaze-contingent window with a target × window presence repeated-measures ANOVA. Since we have already discussed the effect of the target, only window-presence was reported. Window-presence had a significant main effect (F [1, 7] = 9.950, p = 0.016, $\eta_p^2$ = .587). In the post-hoc pairwise comparison with Bonferroni correction, the response time was significantly longer (mean difference: 1,083 ms, p = 0.016) with the window-presence (M = 3,952 ms) than with window-absence (M = 2869 ms). The ANOVA confirmed that the response time was longer with the gaze-contingent display than without a window.

**Table 2. Estimated covariance parameter and fixed coefficients for the authorship questionnaire, comparing the "other" to the "delay" response.**

| Covariance parameter | Estimate | Standard error | Z score | P value |
|---|---|---|---|---|
| Participants | 0.254 | 0.194 | 1.312 | .190 |
| Fixed effects | Coefficient | Standard Error | T score | P value |
| Intercept | -3.903 | 0.325 | -12.021 | .000 |
| Delay | -0.068 | 0.213 | -0.318 | .750 |
| Low visibility vs. high visibility (LV–HV) | -0.088 | 0.296 | -0.296 | .767 |
| Delay × LV–HV | -0.080 | 0.248 | -0.322 | .747 |

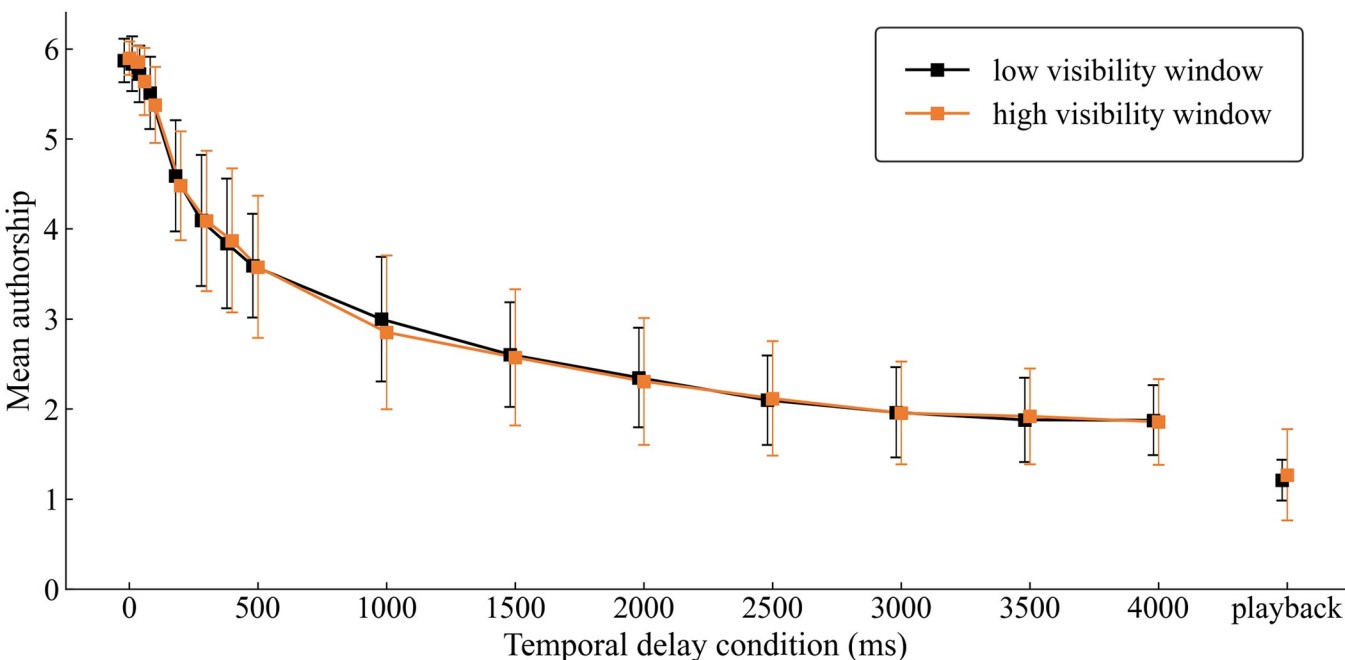

**Fig 4. The authorship rating was a 6-point rating scale of the degree of authorship, ranging from 1 "I could not manipulate the window at all" to 6 "I completely manipulated the window".** Mean authorship ratings are reported by window visibility and delay. Means and standard deviations were computed among the participants. All data points have been offset (dodged) to improve clarity and avoid overlap.

In terms of our aim to find the critical temporal delay, the post-hoc pairwise comparison with Bonferroni correction showed that the response time was significantly different compared to the 0 ms delay (M = 3909 ms) when the delay was 300 ms (M = 6163 ms, p = 0.009) and above (p < .05).

**Eye movements.** The mean relative fixation duration frequencies were calculated among the participants (Fig 6). In the no-window condition, the mode of distribution was located at approximately 125–175 ms, demonstrating gaze behavior similar to that of a visual search [70]. With the gaze-contingent window, the mode shifted to 175–225 ms for the 0 and 30 ms delay conditions. The mode further shifted and exhibited negative skewness as the delay increased. Intriguingly, from 200 ms to 500 ms delay, two modes appeared: one located approximately at 175–225 ms and another which shifted with increasing delay. From 1,000 ms to 4,000 ms delay, including the playback condition, the mode consistently occurred at 150–225 ms. No clear difference was observed between the two window visibility conditions.

The means of the relative saccade amplitude frequencies were calculated (Fig 7). Generally, two distributions were observed across all conditions: one mode was located below 1˚ and another at 7–8˚. The closest distance between the two characters was 8˚, which likely contributed to the peak at 7–8˚. The relative frequency under 1.5˚ appeared to increase with the gaze-contingent window compared to the no-window condition and rose further as the delay increased. The increasing tendency of the relative frequency below 1.5˚ was not consistent in the range of 1,000 ms to 4,000 ms delay in the high visibility condition. Otherwise, no clear differences were observed between the two window visibility conditions.

## Contrast analysis

Following the observation of the results within the 0–500 ms range, we calculated the position on the x-axis of the shifting mode in fixation duration and the cumulative distribution for

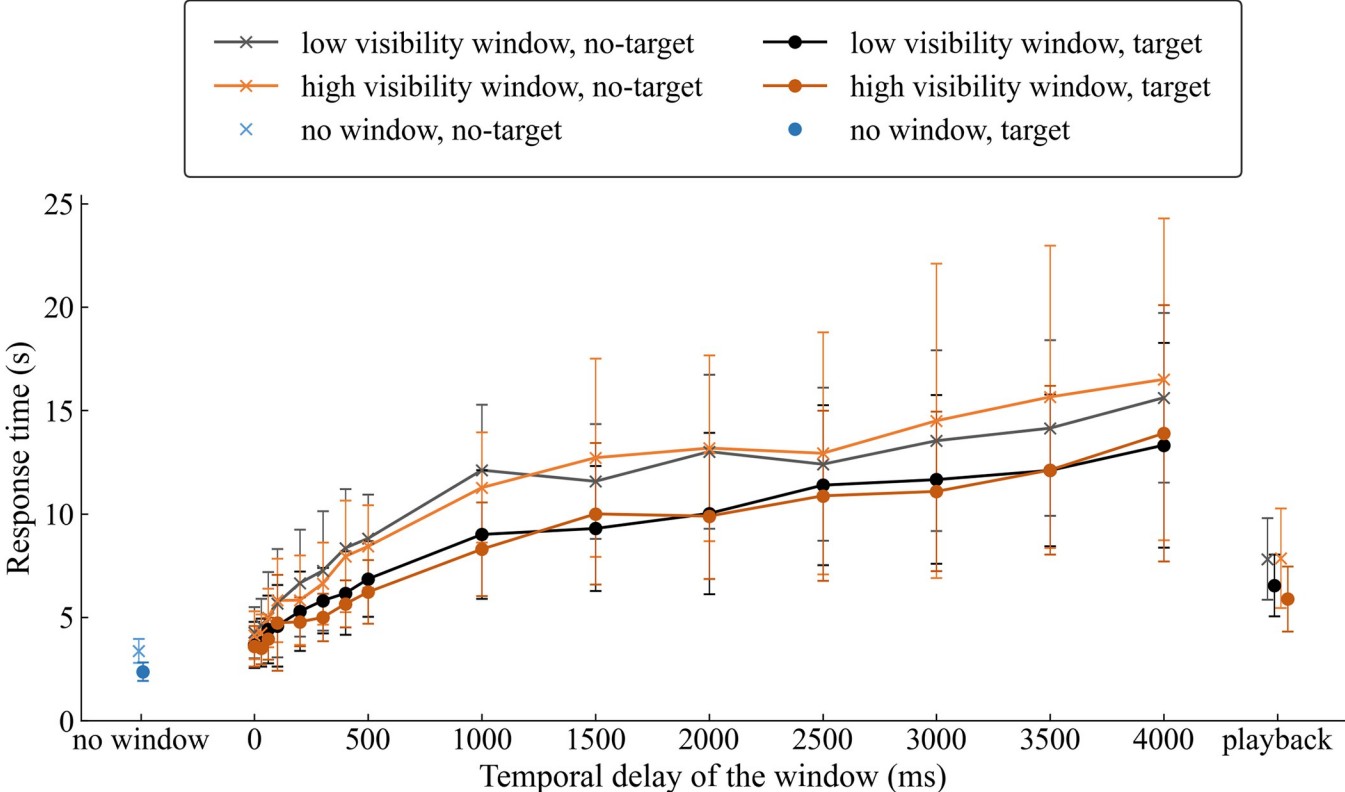

**Fig 5. Means and standard deviations of the participants' response times.** Only the data points for the "no window" and "playback" conditions have been offset (dodged) to improve visibility.

saccades at 1.5 degrees. Additionally, we calculated the amount of change in each value, including the proportion of "Self" responses. Fig 8 presents these results. Since visibility did not exhibit any significant effects in both multivariate ANOVA analyses, we combined the visibility conditions for plotting. Note that we pooled the proportion of "Self" responses across the 0–500 ms range from Fig 3.

Two multivariate ANOVA analyses were conducted: one for delay × visibility and another for delay increase × visibility. For delay × visibility MANOVA, there was a significant main effect of delay (Pillai's Trace = 1.659, F[15, 252] = 20.781, p < 0.001, $\eta_p^2$ = .553). No significant main effect was found for visibility (Pillai's Trace = .037, F[3, 82] = 1.036, p = 0.381, $\eta_p^2$ = .037). The interaction between visibility and delay was not significant (Pillai's Trace = .081, F[15, 252] = .465, p = 0.956, $\eta_p^2$ = .027). For delay increase × visibility MANOVA, there was a significant main effect of delay interval (Pillai's Trace = 1.254, F[12, 210] = 12.571, p < 0.001, $\eta_p^2$ = .418). No significant main effect was found for visibility (Pillai's Trace = .006, F[3, 68] = .145, p = 0.933, $\eta_p^2$ = .006). The interaction between delay interval and visibility was not significant (Pillai's Trace = .191, F[12, 210] = 1.188, p = .293, $\eta_p^2$ = .064).

Pairwise comparisons were performed for both MANOVA analyses. The analyses confirmed that the proportion of "Self" responses decreased, while the fixation duration mode location and cumulative distribution for saccades at 1.5 degrees both increased with delay. The amount of change in the "Self" responses between the 100–0 ms and 200–100 ms intervals was significantly different from the other three comparisons. The changes in the 100–0 ms (M = -.348) and 200–100 ms (M = -.481) intervals were not significantly different from each other.

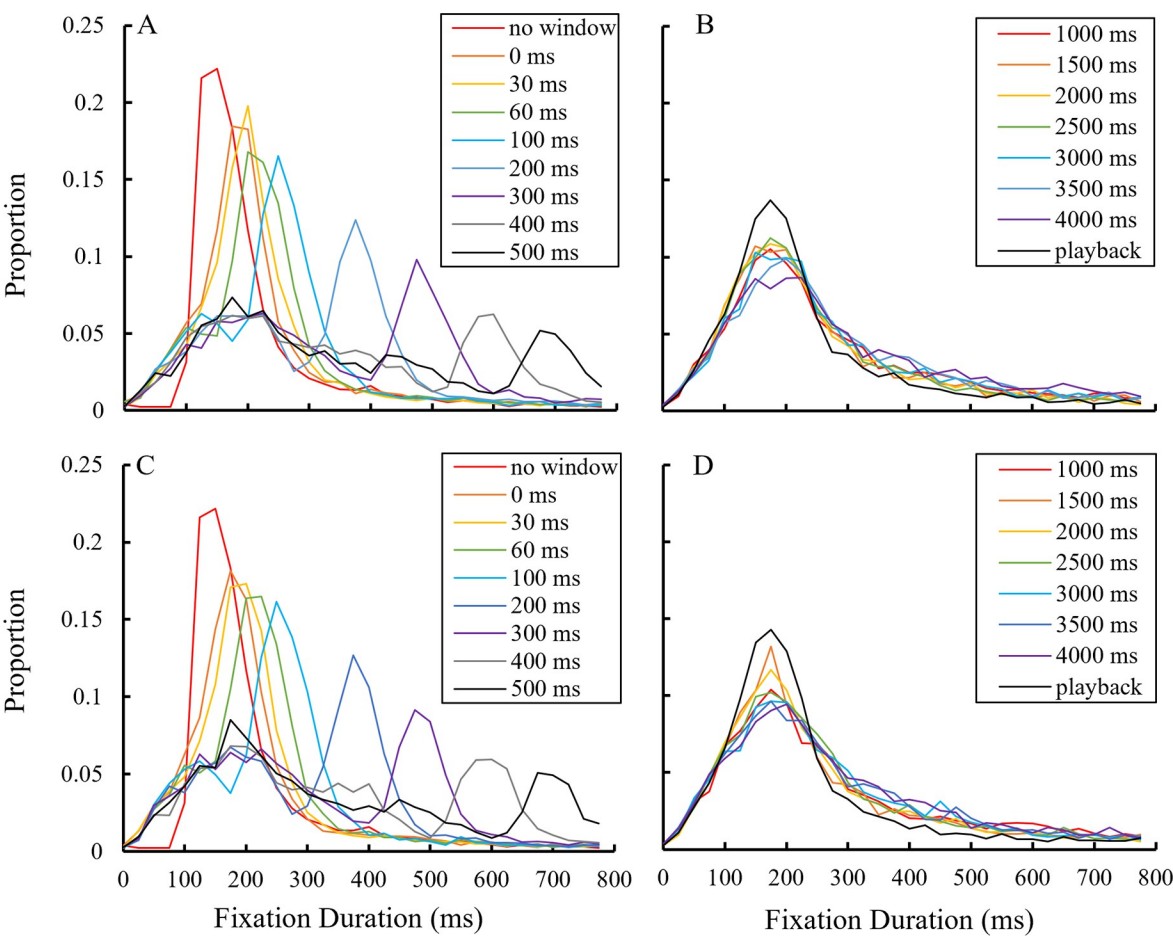

**Fig 6. Relative fixation duration distribution.** The frequency of fixation duration that fell within each 25 ms bin at a 0–800 ms range was calculated for each participant. The graph indicates the means among the participants, with a representative value of each bin placed at the minimum value of that bin (e.g., the value of a 25–50 ms bin was placed at 25 ms on a horizontal axis). (A, B) distributions with a low visibility window. (C, D) distributions with a high visibility window.

Similarly, the changes in the 300–200 ms (M = -.090), 400–300 ms (M = -.010), and 500–400 ms (M = -.006) intervals were not significantly different from each other. These results indicate that the proportion of "Self" responses decreased from 0 to 200 ms, then converged at delays above 300 ms. The fixation duration mode location showed that the amount of change in the 100–0 ms (M = 75 ms) and 200–100 ms (M = 124 ms) conditions was significantly different from the other three compared conditions. Additionally, a significant difference (50 ms, $p < 0.001$) was observed between the 100–0 ms and 200–100 ms conditions. The changes in the 300–200 ms (M = 107 ms), 400–300 ms (M = 101 ms), and 500–400 ms (M = 104 ms) intervals were not significantly different from each other, indicating convergence at delays above 300 ms. The cumulative distribution for saccades at 1.5 degrees showed a significant difference ($p = .003$) only between the 200–100 ms (M = .053) and 500–400 ms (M = .012) intervals.

## Discussion

Here, three main objectives examining the relationships between action-feedback temporal delays, the sense of agency, and eye movement were addressed. First, the categorical question

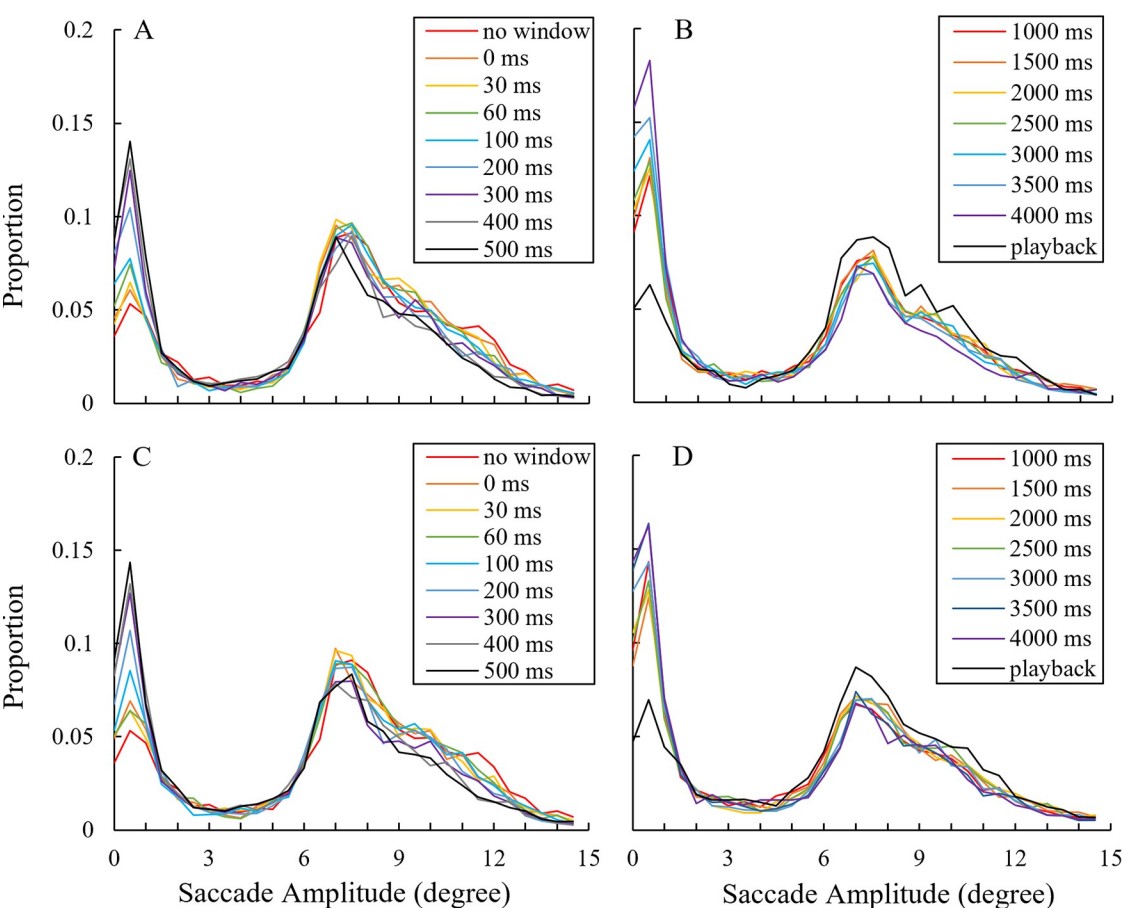

**Fig 7. Relative saccade amplitude distribution.** The frequency of saccade amplitude that fell within each 0.5˚ bin at the 0–15˚ range was calculated for each participant. The graph indicates the means among the participants, with a representative value of each bin placed at the minimum value of that bin (e.g., the value of a 1.5–2˚ bin was placed at 1.5˚ on a horizontal axis). (A, B) distributions with a low visibility window. (C, D) distributions with a high visibility window.

investigated participants' judgment of event authorship with three options (self, delay, other). Second, the rating question assessed the variance in the intensity of subjective authoring. Regarding the first and second objectives, as a side objective, we calculated the critical temporal delay at which participants' experiences began to be affected. And, the third objective explored how temporal discrepancies affected participants' behaviors.

## Sense of agency

In this section, we address the first and second objectives. We hypothesized that the results would mirror those observed in limb modalities: 1) participants would attribute authorship to themselves, not the computer, despite temporal delays in eye-gaze HCI, and 2) the perceived intensity of authorship ('I') would decrease with increasing temporal delays up to 300–500 msec. In this section, we address these two hypotheses.

As expected, participants did not attribute the computer as the event author when they could control the eye-gaze HCI with delay. Using categorical questionnaires, participants consistently distinguished between the delayed and uncontrollable (i.e., playback) conditions by reporting "other" responses only during the playback conditions. According to Synofzik et al.'s [50] framework, the contingency between motor intentions and outcomes is sufficient to

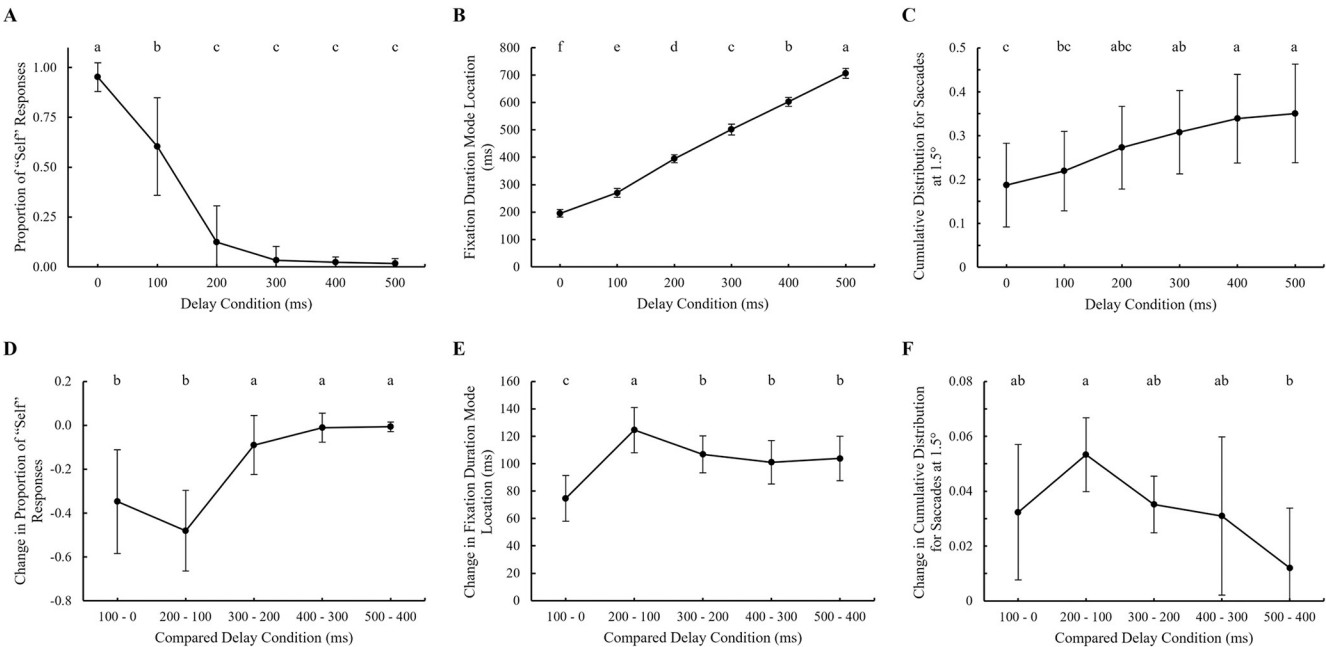

**Fig 8.** The proportion of "Self" responses (A) was re-plotted from Fig 3, with visibility condition combined. The location of the shifting mode in fixation duration (B) and cumulative distribution for saccades at 1.5 degrees (C) are also shown. The amount of change was calculated for each 100 ms increase in delay for "Self" responses (D), fixation duration mode location (E), and cumulative distribution for saccades at 1.5 degrees (F). Note that the visibility condition was combined for plotting as the variable did not exhibit a significant effect. CLD graphs illustrate the pairwise comparison results. Different letters indicate statistically significant differences between delay conditions (p < 0.05), while shared letters signify no significant difference between the conditions.

judge the event author as self. We instructed the participants that the goal of a trial was to visually search for the target character. Thus, participants planned a visual search, likely a serial search [71], as the response time was consistently longer without the target in the stimuli, although it was not twice as long as when the target was present, as was shown in past visual search researches. In the playback condition, regardless of participants' eye movements, the window moved around. In line with Treisman et al. [71]'s feature-integration theory, which explains that visual searches involving shared visual elements (e.g., similar conjunctions of multiple lines, and similar spatial frequencies derived from the same stroke numbers of Chinese characters) require serial search, the playback condition likely disrupted participants' ability to efficiently search for the target, as the window's movement did not align with their planned search strategy. To read a character within the window, participants were forced to discard their saccade plan and strategy and follow the window's movement, which they could not expect within a trial. Also, due to the random sequencing of conditions between trials, participants could not predict whether the window would move independently of their actions. As the window did not correspond with the participants' saccade plan and search strategy in the playback condition, the incongruency between motor intentions and outcomes is considered to account for the reported "other" responses. This finding aligns with prior studies, such as Sato and Yasuda's [21] first experiment, which demonstrated that the sense of agency weakens when there is a mismatch between predicted and actual outcomes. Furthermore, we instructed participants that the computer could control the window. Such contextual cues also play a significant role in forming the response judgment, as per Synofzik et al.'s [50] framework. Regarding to the second objective, the clear distinction in event author inference is also found in the rating questionnaire, which had the lowest scores in the playback condition. The

continuous rating assessed the intensity of influence exerted by the "self." When participants judged the event author to be the computer in the uncontrollable playback conditions, they chose the minimum scale of 1 to indicate the minimal involvement of "self" in the outcome. The dissociation between response time and rating reports in the playback condition makes it unlikely that participants based their ratings on task complexity.

When there was a delay at the gaze-contingent window, participants responded with a "delay" response and did not attribute the event to another author (i.e., the computer). Sense of agency has been reported to be maintained when the temporal action-effect mismatch does not exceed the internal time window of around 200–300 ms [72]. Also, this time window has been suggested to buffer own movement [72]. We assume that two aspects of the experimental action feedback paradigm were key factors in forming the judgment: the number of inputs (continuous vs. single action effect paradigm) and the degree of freedom in action-effect (directing vs. a button push). While "other" responses were not reported in delayed conditions in both the current study and Farrer et al. [22], Farrer et al. [23] showed "other" responses with delays of around 700 ms. As we did not limit the participants' response time, they conducted a trial until they decided to terminate. Indeed, they often conducted a trial for 10 to 20 seconds when the delay was over 1000 ms. During this time, participants could continuously navigate their gaze until they formed the judgment of agency. In Farrer et al. [22], participants could freely direct the joystick to update the virtual hand's position for 5 seconds. Meanwhile, in Farrer et al. [23], participants were only able to click once (a single button press) and the stimuli updated the feedback only once, with the appearance of a grey ball. The amount of acquired sensory information would be largely different between continuously directing and pushing a button only once. This difference in action-effect paradigms also reflects the difference in delay where participants reported a "delay" response. In both the current study and Farrer et al. [22], "delay" responses were dominant from 200 ms. Further, for comparison with Farrer et al. [23] where the inflexion point was reported, both studies' inflection points appeared at 100 ms. Meanwhile, in Farrer et al. [23], the inflexion point was around 330 ms.

The participants continued to attribute control to themselves, even at the longest delays of four seconds. It is implausible that participants' action-effect time window buffers their own multiple numbers of eye movements up to 4 seconds. So, how did participants distinguish the event author as their own in such extended delayed conditions? A practical example of such self-authorship with a prolonged delay can be seen in everyday situations, such as pressing a button on a coffee machine or an elevator. When pressing a button on a coffee machine, even though there is a known delay of about one minute before the coffee drops into the glass carafe, individuals still perceive themselves as the author of the dispensed coffee. Similarly, when pressing the call button for an elevator, people hear an alarm sound of the elevator arriving after several seconds' delay and still perceive their action as the cause. While there is no clear research on how the sense of agency diminishes with delay yet remains robust at prolonged delays (mismatch), one explanation of our findings can be made using the concepts of non-conceptual and conceptual agency (feeling and judgment of agency) as proposed by Synofzik et al. [50]. The decrease in "self" response proportions around 200 ms suggests that the non-conceptual feeling of agency was disrupted due to the mismatch between predicted and perceived feedback. This disruption likely led participants to question their immediate authorship of actions, reflected in their diminished sense of agency. Synofzik et al. [50] suggested that the comparator model could be a mechanism in the disruption of the non-conceptual feeling of agency, and we speculate that the decrease in the "self" response proportions observed in our experiment was due to the internal comparison process described by the comparator model. Our contrast analysis (Fig 8) revealed significant decreases in "self" response proportions from 0 to 200 ms. The "self" response proportions fell near 25% at a delay of 200 ms. Also from 300

ms, "self" response proportions converged to near 0%. While the Synofzik et al.'s framework did not suggest any time window for the comparator model, based on the 200 ms time window reviewed by Wen [72], we speculate that participants consistently perceived the delay above 200–300 ms as exceeding the time window associated with the feeling of agency (referred to as body agency in Wen [72]). Also, we speculate this non-conceptual process can be a part of the "minimal self" described by Gallagher [73]. According to Synofzik et al. [50]'s framework, this perception can be explained by the incongruency between expected states (high-resolution image) at the attended object and actual states (blurred image).

Meanwhile, the persistence of "delay" (delayed-self) responses at longer delays suggests that participants' sense of agency was maintained. Synofzik et al. [50] discussed the phenomenon of "self-despite-mismatch" (p. 223), highlighting the limitations of the comparator model in fully explaining situations where participants continue to attribute authorship either to themselves or to the computer (as in our experiment, where they kept attributing it to themselves). Synofzik et al. [50] suggested that higher-order cognitive processes, such as beliefs and contextual cues, are significant in the judgment of agency. This experiment is fundamentally based on the participants' belief in the computer's system, specifically that the gaze-contingent window would generally follow their eye-gaze's spatial location with various delays and rarely fail to do so (only in the playback condition). Participants did not consider the playback condition as a mere bug in the system but reported it as the computer's agency ("other" response). The belief in the system was shaped by the experiment's instructions and the repetition of sessions. During the practice trials, participants were explicitly told that the system might sometimes control the window independently, reinforcing their belief that the playback condition was part of the experimental design. Thus in the delayed conditions, the participants could plan a visual search, expecting the display to eventually follow their eye movements (refer to the Behavior section for a detailed discussion on their strategies with eye movements), leading participants to report "delay" responses rather than "other" responses. It can be speculated that, despite disruptions in the immediate sensory feedback (i.e., disrupted feeling of agency), higher-order beliefs were sufficient to form the judgment of agency. Also in the sense of agency regarding the social gazing, it was reported that temporal delay alone could not lead to the lowest rating to the agreement regarding the sense of agency [30]. The lowest rating was shown when the virtual avatar did not follow the participant's own gaze [30]. This suggests that the experience of authoring eye-gaze HCI shares similar aspects with the sense of agency in social gazing.

Synofzik et al. [50] suggested that explicit reports only assess the judgment of agency, while Ebert and Wegner [24] proposed that intentional binding measures implicit feelings of agency. Furthermore, Wen et al. [26] demonstrated that action–effect temporal discrepancy affects the explicitly reported degree of agency toward the outcome but not the binding effects. Although previous research suggests that questionnaires mainly capture the judgment of agency, our findings provide insight into how both the feeling and judgment of agency may be reflected in participants' responses under the current experimental paradigm. Even though the questionnaires measure the explicit judgment, the dynamic shifts in the "self" response proportions as delays increased indicate a subtle interplay between the participants' immediate experiences (feeling of agency) and their higher-order beliefs (judgment of agency).

The GLMM analysis of the categorical response revealed that window visibility significantly influenced participants to perceive the delay at relatively shorter durations. In the high visibility condition, the background brightness was reduced compared to the window. The high visibility condition provided more salient features (e.g., motion perception, contrast in brightness, and difference in spatial resolution between the inside and outside of the window) than the low visibility condition (i.e., only spatial resolution) to distinguish whether the window was

updated at their newly fixated position. It might be that these features were processed in a relatively shorter time, leading to early delay detection. However, early delay detection with a high visibility window was not significant in our contrast analysis and the rating questionnaire. The contrast analysis MANOVA considered not only the "self" response but also eye movement behaviors, which might have led to less sensitivity in reaching a significant effect for visibility. Additionally, the rating questionnaire might have had statistical difficulty in detecting fine differences within the delay range from 0 to 200 ms, as rating scores did not converge to specific scores in the delay above 300 ms. The categorical response showed convergence, resulting in similar outcomes across visibility conditions at delays above 300 ms. This would lead to a relatively high sensitivity in achieving significant statistical results in the delay from 0 to 200 ms. Deficits in statistical power due to our small sample size could also be a contributing factor. These fine differences suggest that the difference in spatial resolution inside and outside the window was sufficient to be attended to in detecting the delay, making visibility a trivial effect. Nonetheless, the results provide insights into designing visual aspects in HCI applications.

While the "delay" response became dominant at delays beyond 200–300 ms, indicating convergence in categorical response results, the mean rating score decreased beyond a 300 ms delay. Consistent decreases in continuous scales have been reported throughout numerous previous studies [21, 24–27], suggesting participants' ratings as a general phenomenon across modalities. The score continued to decrease beyond the delay where "delay" responses converged in categorical reports. Furthermore, the ratings converged as the delay approached 4000 ms. One interesting point observed from the rating scores is that the trend in the graph showed a decrease with a concave upward pattern as the delay increased. In other words, the reduction in scores for the same increase in delay gradually declined with delay, suggesting that participants' sensitivity to assessing their own influence on the outcome decreased with delay. One explanation might be that the comparator model's internal comparison contributed to high sensitivity only at short delays. The participants' rating decrease from 0 to 300 ms tended to be relatively steep compared to the score decrease thereafter. Although this concave upward tendency was observed in our study, it has not been explicitly confirmed in prior studies with limb modalities [21, 24–27]. Thus, further discussion is limited, indicating a need for more research to replicate these findings in different modalities and conditions. Nonetheless, the mean rating score hints at participants' judgment of their own influence in authoring the HCI at longer durations, providing information that categorical reports could not capture.

## Behaviors

In this section, we address the third objective. This study demonstrated alterations in eye movements with the introduction of delays. Initially, we assumed that participants would adopt different task strategies in response to these delays. In this discussion, we explain the task strategies adopted by participants as delays varied, based on the observed eye movement results. Further, we discuss our contrast analysis to explore how these changes in eye movements might relate to the sense of agency.

First, we discuss two points related to the strategies observed in the fixation duration distribution (Fig 6): 1) the shifting tendency of the mode in delay conditions of 0–500 ms, and 2) the unimodal distribution in delay conditions of 1,000 ms and above. We propose that two fixation strategies may account for each point. The first strategy, termed the "wait-and-read strategy," can be explained by considering a scenario where participants make a saccade to an arbitrary character on the array during the visual search. After the saccade, participants maintain their gaze at the fixed location until the gaze-contingent window updates at that position. They then observe the Chinese character. Because participants waited for the window to

update, mode shifting due to the temporal delay was observed in the distribution for delays under 500 ms; specifically, the latter mode occurred between 200 and 500 ms (Fig 6). We now turn to the second strategy, termed "pre-registering." Returning to the scenario where participants made a saccade to an arbitrary character during the visual search, we explain their behavior using the "pre-registering" strategy. After the saccade, participants do not wait at the fixated location for the window to update. Instead, they execute another saccade to a different character. When the delay was sufficiently long, participants repeated additional saccades until they perceived that the window had started to update the first and subsequent saccades. Once the window started to update, it followed the track that participants had pre-registered with their gaze since the first saccade. Participants then followed this window to inspect the character within it. The primary difference between these two strategies is that in the "wait-and-read" strategy, participants' gaze remains at the fixated location until the window updates, while in the "pre-registering" strategy, participants continue to make saccades to different locations without waiting for the window to update.

Our assumption about the "pre-registering" strategy was informed by two sources: first, our direct observations during the experiments, and second, our review of the experimental sessions replayed using the EYELINK Data Viewer's playback animation feature (a typical example video of a trial has been uploaded to the Open Science Framework; see the Data Availability section). The strategy was identified by repeated eye movements along the same trajectory. For instance, participants moved their gaze clockwise or counterclockwise in delays of 1,000 ms and above. For delays below 1,000 ms, they rapidly alternated saccades between two characters. The upward slope of the response time function at delays over 1,000 ms decreased (Fig 5), which supports the strategy change. We assumed that this "pre-registering" strategy manifested as a distribution with a mode located at approximately 200 ms of fixation duration, as observed in Fig 6 under the following delay conditions: 1) the earlier mode from 200 to 500 ms delay, and 2) the mode from 1,000 to 4,000 ms delay. The 200 ms likely reflects the time needed to memorize an arbitrary Chinese character and its location, corresponding to the mode for the non-delayed window ("0 ms" in Fig 6). While the "pre-registering" strategy appeared exclusively when the temporal delay was 1,000 ms and above, the "wait and read" and the "pre-registering" strategies might have occurred simultaneously in the temporal delay condition from 200 to 500 ms with a bimodal distribution.

From the above considerations, two key discussions emerge: 1) The transition from the "wait and read" to the "pre-registering" strategy as temporal delay increased, and 2) The shifted mode in fixation duration distribution. First, participants seemed to adopt a strategy of examining all the Chinese characters in the shortest time. We did not ask the participants to execute articulatory suppression to reduce the verbal encoding strategy. Therefore the phonological loop can be adopted to encode the visual search target character in the center of the display. We obtained participants' verbal reports after the experiment on this strategy. Participants were asked to report the characters that they could read. They reported 1–3 characters out of 239 in our stimuli, all representing specific fish species and having extremely low familiarity and appearance rates in our daily lives. Hence, it can be assumed that it was not easy to rely on the verbal encoding strategy for the participants to keep the target in memory during the task. Participants had to consider a strategy other than "wait and read" at extended temporal delays, to inspect as many surrounding characters as possible in the shortest time before the visual memory persistence for the central character is lost. The first author's subjective impression as a participant was that rather than backward masking from each fixation, it was simply related to the duration of the memory decay after the inspection of a character. Consequently, they had to employ the "pre-registering" strategy in extended temporal delays, such as those longer than 1,000 ms, to avoid the image lost to execute the task. This transition in strategy would be

explained without incorporating the framework of the sense of agency. The downward slope of the rating response decreased over 1,000 ms of delays (Fig 4), suggesting a decrease in participants' sensitivity to assessing their own influence on the results in delays over 1,000 ms.

Second, we address the mode shift. Our data confirmed that the proportion of "self" responses decreased in the delay range from 0 to 500 ms. To discuss how the sense of agency might be related, we focused on how the shifting mode was influenced by the delay in this range. Contrast analysis revealed that the mode shift was significantly longest from 100 to 200 ms. This delay range coincided with a significant decrease in "self" responses, falling to about 25% at 200 ms. This suggests that participants' perception of delay might contribute to the radical shift in the shifting mode. The comparator model might explain this, as the bodily agency is reported to have a 200 ms time window [72]. The term "wait-and-read" implies that the mode should shift 100 ms for every 100 ms increase in delay. This pattern was observed in the 300 to 500 ms range. Interestingly, the shift from 0 to 100 ms was shorter than other shifts in the 300 to 500 ms range, while the shift from 100 to 200 ms was longer. The short mode shift from 0 to 100 ms likely contributed to the longer shift from 100 to 200 ms. The delay from 0 to 100 ms was where the "self" proportion significantly decreased to 50%. According to the comparator model, it is plausible that participants' action refinement was suppressed when the bodily agency was maintained, as the temporal mismatch did not exceed the time window, then amplified when the mismatch was detected at 200 ms.

Building on the observed strategy shift and mode transitions in response to delays, we now focus on how these strategies specifically reflect the interaction within eye-gaze HCI. The "wait-and-read" strategy aligns with the real-time, perceptual level of control that is central to immersive eye-gaze HCI. In this mode, participants continuously monitor visual feedback and adjust their actions accordingly, maintaining a direct connection between eye movements and interface responses. However, as delays extend, this seamless interaction breaks down, leading to a shift toward the "pre-registering" strategy, where participants preemptively plan their gaze shifts based on perceived delay. We suggest that this shift reflects a transition from perceptual agency to belief-based judgment, a move from immediate feedback reliance to a premeditated mode of control. The "pre-registering" strategy involves a loss of spontaneous, first-person interaction, as users adapt to the prolonged delays by relying on memory and prediction, undermining the seamless and intuitive experience expected in continuous eye-gaze HCI. Such transitions highlight the critical role of minimizing delays to preserve the immediacy and fluidity of eye-gaze interactions.

Upon examining the saccade amplitude distribution, we observed an increased relative frequency of 0–1.5 degree saccades as the temporal delay increased (Fig 7). Although we are skeptical about the camera-based eye-tracking's ability to capture microsaccades, we can infer the underlying causes by drawing insights from previous research on microsaccades. Privitera et al. [74] found that microsaccade frequency increased during visual searches involving ambiguous targets. Microsaccades also help obtain spatial detail [75] and reposition the gaze in tasks requiring fine spatial discrimination [76]. Another study showed that participants could voluntarily generate microsaccades when re-fixating on a memorized target [77]. Studies have shown that microsaccades are suppressed as the expected time of target appearance approaches [78–80]. Also, pupil diameter stabilizes when a target is anticipated [81], with fluctuations subtly affecting gaze position as tracked by eye-tracking devices [82]. Microsaccades are considered proxies for measuring covert attention allocation [83, 84]. They also become less frequent as task difficulty increases in both non-visual [85] and visual tasks [86].

While the Chinese characters used in this experiment required participants to acquire fine spatial details when discriminating characters with partial shapes, the increasing trend in saccades under 1.5 degrees with delays is difficult to explain. Task difficulty is an unlikely

explanation, as previous studies reported a decreasing trend in these saccades with increased task difficulty. A more plausible explanation involves the impact of delays on both temporal and spatial attention. Regarding temporal attention, immediately after a saccade, participants encountered an unexpected blur instead of the anticipated clear character, disrupting their ability to predict when the gaze-contingent window would align with their central vision. This led to a decreased reported sense of agency and likely disrupted microsaccade suppression, aligning with Dankner et al. [80], who found that microsaccades are inhibited when anticipating a predictable stimulus. Additionally, the delay may have caused a detachment between eye-gaze and attention allocation, creating a spatial distance between the participants' gaze and the attended window. As this distance increased with the delay, so did the covert attention to the window, likely contributing to the observed increase in smaller saccades.

How could the increase in saccades under 1.5 degrees relate to the sense of agency? We conducted contrast analysis at the cumulative distribution at 1.5 degrees. While it was not statistically significant, the increase was the largest in the delay range from 100 to 200 ms. This suggests that participants' attention distribution might be modulated with delay perception in this range. We propose that the spatial and temporal attention distribution might be part of the internal mismatch comparator, given that attention is necessary for comparing action and its effect [87]. Also, Taylor [88] proposed that corollary discharge (i.e., efference copy at eye movements) is generated from an attention-shifting signal rather than a motor signal, and suggested that our consciousness derives internal cues from this signal.

## Critical temporal delay

As a side objective derived from the first and second objectives, we examined the critical temporal delay at which user experience was interrupted. Post-hoc analysis identified the critical temporal delay as 300 ms for response time, 200 ms for the categorical authorship report, and 200 ms for the mean authorship rating. The critical delay of 200 ms overlayed with our results of contrast analysis where categorical authorship report and fixation duration showed significant changes. As we speculated in the discussion of contrast analysis, it might indicate the time window at the feeling of agency [72], suggesting the relation between the alteration of user experience and bodily agency. The current results may reflect attempts to reallocate attention after the disruption in prediction, as participants struggle to re-engage with the task. Kamienkowski et al. [89], who uniquely tested rapid serial visual presentation (RSVP) under saccadic conditions, found that the attentional blink occurred 200–500 ms after the first target, even when saccades were made. While the time window resembles the window in the bodily agency, the relation between RSVP, attentional blink, and sense of agency remains an open question.

Our results showed inconsistency between subject reports and response time. Although different measures might have varying intrinsic sensitivities in detecting delays, the observed inconsistency suggests a power deficit in our analysis, which was limited to eight participants. However, this power deficit does not conclusively rule out the possibility of influences at shorter delays. Inspection of the proportion of categorical authorship responses, fixation duration distribution, and saccade amplitude distribution revealed a tendency to deviate from the 30 ms delay, suggesting an early effect on performance. Further research with a larger participant pool might reveal a critical delay shorter than the 50–70 ms proposed by prior studies utilizing a 3D graphic scene with a head-mounted display [53] and a natural scene with a 2D monitor [54].

An interesting inspection was that when the temporal delay was further extended beyond the critical delay, despite the fact that users' performance was clearly disturbed, they were able to manipulate the gaze-contingent window and accomplish the task, even at 4,000 ms, as in the

satellite teleoperation scenario where participants teleoperated a robotic arm with their hands under a 7-second temporal delay [47]. However, further study with different experimental designs is necessary to discuss users' delay tolerance in completing tasks, in that our experiment did not limit the task time. Manipulating the number of Chinese characters at stimuli would be another aspect that could be addressed.

## Gaze-contingent display

In this section, we describe the assessment of the stimuli used in the experiment. In the present study, we implemented a 3 × 3 virtual grid of Chinese characters with a gaze-contingent window, aiming to address our main questions while preventing a decrease in saccade amplitude with peripheral blur, as reported in previous studies [61, 63–65, 68]. The results of the present study showed that the gaze-contingent windows increased response time and affected gaze behavior. The fixation duration was longer with the window than without the window, with the mode of distribution shifting from 125–175 ms to 175–225 ms. In terms of saccade amplitude, a frequency under 1.5˚ was more prevalent with the window. Nonetheless, saccade amplitude did not appear to decrease, as 7–8˚ saccades consistently appeared in our results across all conditions, due to the 3 x 3 virtual grid. Overall, while the window influenced participant behaviors, it did not impede our ability to address our questions in this study.

## Study limitations and future directions

This study has several limitations. First, the sample size was relatively small, leading to underpowered analysis when examining critical temporal delay, and the analysis was also limited to a specific age group. Second, explicit questionnaires may not adequately assess the sense of agency, as subjective measures like rating scales and response criteria have intrinsic limitations and are subject to systematic biases. Third, altered eye movements in temporally delayed gaze-contingent displays may not generalize to other stimuli (e.g., natural scenes).

While the intentional binding paradigm has been suggested to measure the feeling of agency, it may not be necessary to rely on this approach for all contexts. Instead, future research could explore a belief-based framework, focusing on scenarios where participants are not provided with explicit cues about who or what is controlling the interface. For instance, a potential experimental design could involve subtly mixing externally controlled movements with participants' own movements without their knowledge. By masking the control source, this approach would allow for an investigation into the sense of agency in the absence of explicit belief or expectation. Such a design could reveal whether participants can still distinguish between self-generated and externally generated actions purely based on their sensory feedback. This would provide a deeper understanding of the processes that contribute to the feeling of agency and could inform the development of more adaptive human-computer interfaces that adjust based on implicit user experience rather than explicit beliefs.

Collecting objective measures using signal detection frameworks could be a valuable direction. For example, Wen et al. [90] introduced a two-alternative forced-choice (2AFC) task where participants identified which of two objects was more controlled by their movements, without receiving explicit cues about control sources. This design relies purely on sensory detection, aligning with approaches that minimize the influence of pre-existing beliefs or expectations. Control was adjusted by varying the proportion of participant-driven versus pre-recorded movements, maintaining 70% accuracy via a staircase procedure. This method offers bias-resistant measures like perceptual sensitivity (d'), providing a reliable gauge of the sense of agency.

## Conclusion

Our study provides insights into how temporally delayed gaze-contingent displays affect the sense of agency and eye movements. By expanding research beyond traditional effector body parts to include eye movements, our findings reveal a complex interplay between delay perception and action refinement. This study offers a novel examination of the relationship between temporal delay and continuous eye movements in eye-gaze HCI, contributing valuable knowledge to the development of delay-tolerant interfaces. Our results also underscore the dual-layered nature of the sense of agency—both the immediate, non-conceptual feeling of agency and the higher-order, belief-based judgment of agency—operating simultaneously under delayed conditions. Even though questionnaires primarily assess the judgment of agency, our findings demonstrate that subtle shifts in "self" response proportions with increasing delays reflect the interaction between both levels of agency. This dual measurement approach offers a more comprehensive understanding of how delays impact the user experience in gaze-contingent systems and lays the groundwork for designing more effective interfaces that maintain a seamless and intuitive experience, even in delay-prone settings.

## Supporting information

**S1 File. This file contains four Appendices: S1-S4.**
(DOCX)

## Acknowledgments

We thank Eric Redlinger (Tokyo Institute of Technology) for his comments regarding this study.

## Author Contributions

**Conceptualization:** Junhui Kim, Takako Yoshida.

**Data curation:** Junhui Kim, Takako Yoshida.

**Formal analysis:** Junhui Kim, Takako Yoshida.

**Funding acquisition:** Junhui Kim, Takako Yoshida.

**Investigation:** Junhui Kim.

**Methodology:** Junhui Kim, Takako Yoshida.

**Project administration:** Junhui Kim, Takako Yoshida.

**Resources:** Junhui Kim, Takako Yoshida.

**Software:** Junhui Kim.

**Supervision:** Junhui Kim, Takako Yoshida.

**Validation:** Junhui Kim, Takako Yoshida.

**Visualization:** Junhui Kim, Takako Yoshida.

**Writing – original draft:** Junhui Kim.

**Writing – review & editing:** Junhui Kim, Takako Yoshida.

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
