## [Decision Letter · Decision Letter 0]

12 Dec 2023

PONE-D-23-32262Sense of agency at a temporally-delayed gaze-contingent displayPLOS ONE

Dear Dr. Kim,

Thank you for submitting your manuscript to PLOS ONE. After careful consideration, we feel that it has merit but does not fully meet PLOS ONE’s publication criteria as it currently stands. Therefore, we invite you to submit a revised version of the manuscript that addresses the points raised during the review process. Editor comment. Two expert reviewers have commented on your manuscript, with one providing a positive review and the other a negative one. Both referees have offered useful comments that merit consideration. A recurring theme raised by both is the introduction of theoretical concepts and mechanisms, as well as the clarity of these concepts and their relationships to the operationalisation in your experiment. Reviewer 2 additionally raises several issues regarding your theorising on efferent-afferent signal discrepancy, arguing that your theoretical explanation is excessively elaborate, containing unnecessary elements that are somewhat platitudinous. In light of these comments, I advise you to address all these points in great detail. Should you feel capable of dealing with the issues raised, I recommend preparing a revised version of your manuscript accordingly.

We look forward to receiving your revised manuscript.

Kind regards,

Michael B. Steinborn, PhD

Section Editor

PLOS ONE

Journal Requirements:

"Takako Yoshida was supported by a ROBOT Industrial Basic Technology Collaborative Innovation Partnership grant from the New Energy and Industrial Technology Development Organization of Japan (Grant number  JPNP20016). The funders had no role in study design, data collection and analysis, decision to publish, or preparation of the manuscript.

https://www.nedo.go.jp/english/activities/activities_ZZJP_100188.html

Junhui Kim was supported by the Japan Science and Technology Agency’s Support for Pioneering Initiated by the Next Generation (SPRING) Program (Grant Number JPMJSP2106).The funders had no role in study design, data collection and analysis, decision to publish, or preparation of the manuscript.

https://www.jst.go.jp/jisedai/en/"

Reviewers' comments:

Reviewer's Responses to Questions

**Comments to the Author**

1. Is the manuscript technically sound, and do the data support the conclusions?

Reviewer #1: Yes

Reviewer #2: No

2. Has the statistical analysis been performed appropriately and rigorously? 

Reviewer #1: Yes

Reviewer #2: No

3. Have the authors made all data underlying the findings in their manuscript fully available?

Reviewer #1: Yes

Reviewer #2: Yes

4. Is the manuscript presented in an intelligible fashion and written in standard English?

Reviewer #1: Yes

Reviewer #2: No

5. Review Comments to the Author

Reviewer #1: The authors investigated the effect of temporal discrepancy between action-effects and explicit measures of sense of agency using eye movements. The results showed that authorship ratings declined as the temporal discrepancy increased and that eye movement behaviour changed as a result of the discrepancy.

I thought that this study was interesting and I appreciated the several real-world examples that were given such as the remote surgery. Overall, I believe that this study should be published. At the same time, I feel that there are several loose ends that could be addressed in a revision:

1. In the introduction, could you please discuss in further detail the active rubber hand illusion and intentional binding paradigms? As it currently stands, it may be difficult for the reader to understand what those experiments involved. For intentional binding, you may also want to make clear that the Libet clock is only one of several methods of assessing intentional binding.

2. In the introduction, when discussing the comparator model, you may also want to mention the sensory attenuation phenomenon. Nathan Mifsud, for example, has looked at sensory attenuation in the context of saccade-initiated stimuli.

3. In the Methods section, you first detail the number of participants in the study, but it takes some time before you mention the number of trials then some more time before you mention the number of sessions. I suggest that you mention earlier the number of trials/sessions. Although you highlight the underpowered nature of the study at the end, stating the number of trials/sessions earlier may help to put into context the 8 participants that you end up using.

4. You asked the participants if they could remember the stimuli to evaluate if there were any effects based on sequence repetition. In principle, you could also compare the response times between the first and second sequences.

5. Under “Eye movements” (line 398), could you please specify how you arrived at the thresholds for saccade detection?

Reviewer #2: The authors claim to investigate the sense of agency using temporally delayed gaze-contingent display. Although the analyses and the results seem appropriate to some degree, unfortunately, it is difficult to know what the purpose of those analyses and results are, and therefore I cannot determine whether the manuscript as a whole is a scientifically rigorous study. The entire manuscript needs to be thoroughly reorganized and rewritten. See below for detailed comments.

1) The manuscript proceeds without first presenting a hypothesis; however, it is concluded that the hypothesis is supported. Specifically, in the Introduction, the authors describe the previous studies and state that the question of this study is to examine whether sense of agency is affected by temporal delays in eye-gaze input devices (l.125). Nevertheless, in the Discussion, they state that “the first question aimed to investigate how the efferent–afferent temporal discrepancy influences the sense of agency during continuous eye movements and stimulus updates (l.464-465). If this manuscript aims to test the hypothesis that a time delay causes a temporal discrepancy of afferent and efferent signals, which in turn reduces the sense of agency, please state that in the Introduction.

2) The manuscript does not show what psychological constructs correspond to what scales and responses, and thus what hypotheses are supported by the results. If my guess is correct, the authors seem to suggest a relationship between the discrepancy between efferent and afferent signals and the sense of agency, because two different scales yielded different result. However, the authors do not state how the variation of these two scales would suggest which of the presented constructs (e.g., efferent-afferent signal discrepancy, action-effect incongruence, non-conceptual feeling of agency, conceptual explicit conceptual judgment of agency, and arousal) are at work.

3) The fact that there is a minute (continuous) change in the sensation that participants are moving display, even though they at least know the fact that they are moving it, does not seem to support the underlying function of the efferent-afferent signal discrepancy (l.474-487). To my eyes, it appears that the time delay of the screen following their gaze was not simply long enough to make them think they were not moving the screen at all, but long enough to produce minute (continuous) changes in the loss of that sensation. If we were to follow Occam's razor, wouldn't there be no need to present the additional concepts of efferent and afferent signals? The same is true for the interpretation of eye movements. There should be no need to further add signal discrepancy concepts even though the results are already explained by the two strategies.

4) The sample size of 8 is very small compared to what is typical in the cognitive psychology field and raises concerns about the results.

5) In terms of writing, sections and paragraphs are not well organized and difficult to read.

For example, at the beginning of the Method, instead of giving an overview, the authors give the methodological details and the reasons for adopting such a procedure. In my opinion, important information scattered in the subsections included in the Method (e.g., l.224-226 and 233-235) should be summarized and described here. Note that this also leads to the placement of Figure 1 caption in a position that does not follow the PLOS guideline because it is not inserted immediately after the first place mentioned in the text.

Also, I think that readers would like to know what the questionnaire captures and how the authors expect it to generate a general replication of past results (l.157-158). In the current manuscript, only the former is revealed later (l.277-283).

Furthermore, l.284-292 are the first information that should be written in Conditions and Procedure section. I was forced to read the manuscript not knowing if the conditions were random or fixed during a certain session. Also, if the key condition is the difference in time delay, then there is no need to itemize and highlight the conditions about presence or absence of window, which only provide control condition (l.251-258).

6) Often there is no explanation of important information while there are many unimportant explanations. While it is good for reproducibility to describe procedures in as much detail as possible, too much detail can reduce the readability of the manuscript. Examples are listed below.

Why would participants be able to detect efferent-afferent temporal discrepancy by observing changes in the brightness of the boundary (l.136-137)? What this observation allows participants to do is to become more sensitive to the external stimulus feedback that is occurring in correlation with their eye movements. Unless you state that prior research has shown that such feedback exclusively reflects an afferent-efferent discrepancy, you are making a leap of logic.

Some sentences (l.180 to 181 and 191) do not seem necessary. It should be sufficient to simply state that the test was performed in a dimly lit room so that key press responses are possible.

What is the meaning of l.210-213?

What is the meaning of (B) in l.246 and (C) in l.248?

I don't understand what l.272-273 are explaining, and it would be better to explain first what the objective of the playback trials is.

If the screen refresh rate is 144 Hz for 2000 Hz eye movement measurements, it should be necessary to approximate the eye movements in time, but the method is not described. Note that eye movements are very fast, and the temporal resolution is critical.

Why does pop-out occur in the letter category (l.208)?　According to Triesman's feature integration theory, pop-outs occur on screens where simple features such as color and shape are uniquely different.

7) I don't particularly understand what the purpose of the first analysis (l.301-) was. What is the purpose of showing the task difficulty? Furthermore, I can't say with confidence that reaction time reflects difficulty because participants are asked for a two-step response and given no instruction to respond as quickly as possible (l.153-154).

9) In l.358-362, the authors state that the lack of statistical significance is due to the small sample size, but immediately afterwards they conclude that there is no difference because there is no significant difference, which is contradictory. Furthermore, the absence of a statistically significant difference does not directly indicate the absence of a difference.

I hope that these comments will help you to improve future manuscripts.. Thank you for the chance to review your research.

6. PLOS authors have the option to publish the peer review history of their article (what does this mean?). If published, this will include your full peer review and any attached files.

Reviewer #1: No

Reviewer #2: No

---

## [Author Response · Author response to Decision Letter 0]

14 Feb 2024

13-Feb-2024

Michael B. Steinborn 

Section Editor

PLOS ONE

Dear Editor:

We wish to re-submit the manuscript titled “Sense of agency at a temporally-delayed gaze-contingent display.” The manuscript ID is PONE-D-23-32262.

We would like to express our sincere gratitude to the editor and reviewers for their valuable time and effort in reviewing our paper and providing constructive feedback that has significantly contributed to the improvement of our manuscript. We are also grateful for the granted extension, allowing us to further refine our work. We have made the necessary changes according to the feedback provided. We believe that these revisions have enhanced the quality of our work. Detailed responses addressing the concerns raised by the editor and reviewers can be found below. 

Response to the comments from reviewer 1:

1. We added a detailed explanation for the experimental setup of the active rubber hand illusion in lines 64-71. For intentional binding, we introduced a brief review of the phenomenon and addressed your recommendation to comment that the Libet clock is one of several methods of measurement (lines 71-79).

2. The sensory attenuation phenomena in the visual domain have been mentioned concerning the central cancellation where we introduced the comparator model (lines 50-54).

3. We have introduced the number of trials and sessions earlier in the Methods section, now appearing on line 202. The substantial time required for the study (30 sessions requiring 5-6 consecutive hours) significantly limited our ability to recruit additional participants. Despite efforts to increase participant numbers following the initial decision letter, these recruitment attempts were unsuccessful. The impact of having a lower number of participants has been stated in the limitations section of our manuscript.

4. Investigating the effect of stimuli sequence presents a challenge due to its close association with session sequence. Each stimuli sequence encompasses 30 sessions, resulting in a significant overlap in these variables. This is evidenced by moderate multicollinearity (session sequence: Tolerance = .25, VIF = 4.005; stimuli sequence: Tolerance = .25, VIF = 4.005). Furthermore, in our generalized linear mixed model analysis, which included saliency, delay, target presence, session sequence, and stimuli sequence as fixed effects (with participants as random effects). The fixed effect for stimuli sequence could not be computed, resulting in “zero degrees of freedom.” Significant fixed effects were found for delay, target presence, and session sequence, while saliency did not show a significant effect. We implemented counterbalancing of stimuli sequence across participants to mitigate this issue, particularly when analyzing response time with the independent variable “saliency condition.” 

5. It was pre-defined values at the software and mentioned in lines 451-453.

Response to the comments from Reviewer 2:

Reference to the initial manuscript is indicated by adding "(original)" next to the line references.

1. In this revision, we have clarified our hypothesis (revised lines 125-129) and replaced the term “efferent-afferent” with “action-feedback.” The modification was made in response to the third comment about Occam's razor, as our experimental design does not support an in-depth exploration of efferent and afferent signals.

Furthermore, revisions were made throughout the manuscript, while retaining references to “efferent-afferent” in the introduction when discussing the comparator model (revised lines 43-60 and 92), and in the reason that supports our hypothesis (revised line 131). We revised the section highlighting the research gap about implementing temporal delays in studying the sense of agency with eye movements (from original lines 95-96 to revised lines 101-105). 

We have updated our rationale in the saliency condition in lines 195-199 (revised), initially related to efferent-afferent signals at lines 136-137 (original), to better reflect our methodological approach to saliency variations. This includes enhancing the generalizability of our HCI findings (response corresponds to the sixth comment). We renamed the more/less saliency condition to low/high visibility throughout our manuscript.

In the discussion section, we have replaced the term “efferent-afferent” with “temporal discrepancy” or "temporal delays” (revised lines 493-499). Discussion of the results of the two questionnaires in relation to efferent-afferent concepts, has been revised and is now presented in lines 500-514 of the revised manuscript.

The focus of the discussion on eye movements has been revised from examining the influence of efferent-afferent discrepancy to exploring the impact of temporal delays. This revision is reflected in the shift from lines 511-515 in the original manuscript to lines 538-540 in the revised version.

Additionally, we removed our assumption about how the efferent-afferent discrepancy computation affected fixation duration. This helped streamline the corresponding paragraph from lines 541-560 (original) to lines 566-577 (revised). The discussion of saliency (now visibility condition) with “efferent-afferent” was removed (original lines 613-614), streamlining the last discussion paragraph to lines 627-637 in revision.

Finally, the "systematic examination of the impact of afferent-efferent incongruency" in the conclusion section was revised (original lines 629-640 to revised lines 642-648) and updated in the abstract (line 27 in the original and revised manuscripts).

2. In the revised manuscript, we have clarified that the two questionnaires were posited to capture the explicit judgment of agency, as detailed in lines 173-174. To further elaborate, we added a new paragraph (lines 137-148) explaining the rationale behind adopting these questionnaires and our expectations from them. This addition is particularly relevant considering our discussion on the criticisms of the comparator model by Synofzik et al. (2008) [reference number 29 in the manuscript]. We have specifically addressed the limitations and the utility of our measurement approach in testing our hypothesis (lines 144-148). It is crucial to note that we did not assert in the discussion or conclusion sections that the results from our questionnaires directly support the judgment of agency. Our focus of implementing the two questionnaires was with the capability to test our hypothesis (i.e., measure the impact of delay on the sense of agency, particularly with eye movements.) Our study did not include intentional binding, which has been proposed as a measure of the implicit feeling of agency (lines 93-98). As a result, since we could not compare intentional binding with subjective reporting as done by Wen et al. (2015) [reference number 26 in the manuscript], we refrained from engaging in a detailed discussion on the sense of agency, especially in terms of judgment of agency.

To clearly highlight the distinctions between the two questionnaires, we have included two paragraphs in the manuscript, specifically detailing what each questionnaire measures (lines 149-174).

In the manuscript, we posited a hypothesis that participants' behaviors, including eye movements and response time, would alter their strategies. This hypothesis aligns with the concept of action refinement in the comparator model (lines 175-177). However, due to the exclusion of efferent-afferent concepts in our revised manuscript, we have narrowed our discussion to focus solely on the impact of delays on these behavioral strategies.

3. (In response to the fourth comment) We have contributed to expanding previous findings by introducing eye movements as a new modality to explore the impact of action-feedback delays on the sense of agency. Additionally, our findings assist in applying eye-gaze HCI across various delay scenarios, including extreme cases like teleoperation, and in diverse saliency settings. While we acknowledge that the small sample size is a limitation, it does not significantly detract from the value of our study or its suitability for publication. The small sample size limited our ability to finely dissect the “critical temporal delay” and conduct detailed statistical testing on eye movements. However, the clear trends observed in relation to delays and alterations in eye movements substantiate our claim of being the first systematic investigation into how delays affect task strategies in this context.

As previously mentioned in response to Reviewer #1, the extensive time commitment required for participation (30 sessions of 5-6 consecutive hours each) was a major factor in limiting our ability to recruit more participants. Despite additional recruitment efforts following the initial decision letter, we were unable to increase our participant pool.

We have highlighted the issue of the small sample size in the limitations section (lines 650-652) of our manuscript. We trust that readers will appreciate the context and constraints under which our study was conducted, understanding the implications of the sample size on our findings. 

4. In response to the fifth comment regarding the organization and clarity of our manuscript:

Method Section Overview: We have restructured the beginning of the Method section to provide a clearer overview. The number of sessions/trials and the presentation of delayed trials are now introduced first (lines 202-217). Additionally, the “playback” and “no window” conditions are explained in this paragraph. The rationale for the playback condition detailed in the introduction as an uncontrollable condition (lines 159-161).

Reorganization of Content: The details previously found in lines 224-226 and 233-235 of the original manuscript are now concisely summarized in lines 227-228 and 225-227 of the revised manuscript, respectively. We have ensured that these details are maintained in the revised manuscript (lines 312-314 and 321-324).

Questionnaire Explanation: In the Introduction section we have clarified what the questionnaires capture and how we expect the results to align with previous studies (lines 149-174).

Conditions and Procedure Section: The information initially found in lines 284-292 of the original manuscript has been moved to the beginning of the Method section (lines 202-223). The “no window” condition, itemized in the original manuscript (lines 251-258), has been relocated and described in lines 212-215 of the revised manuscript.

5. In response to the sixth comment:

We have revised the rationale behind the saliency condition in our study, as detailed in the previous sections of this response. Consequently, we have removed the corresponding sentence from the original manuscript (lines 136-137).

We have removed the sentences identified as unnecessary (originally lines 180-181 and 191).

The ambiguity in the original lines 210-213 has been clarified. We revised the sentence to: “The uncertainty was introduced to discourage participants from predicting the target's location before completing the visual search and potentially affecting their behavior.” (revised lines 298-303).

The meanings of (B) and (C) in the original lines 246 and 248, referring to sections in Figure 1B and Figure 1C, respectively, have been made clearer in the revised manuscript (lines 294 and 295).

The rationale behind the 4680 ms mask time, originally mentioned in lines 272-273, has been moved and elaborated upon in lines 332-335.

Regarding the screen refresh rate and eye movement measurements, we have added an explanation in lines 272-276, stating that our software acquires eye movement data and updates internal computation every 1 ms. This was verified through output debugging messages generated in each loop of the computation process.

Our initial assumption posited that characters with fewer strokes would “pop out” when presented in parallel with those having a higher stroke count, owing to the shape differences. Additionally, we considered that characters with different radical elements might also produce a more significant “pop out” effect when combined with varying stroke counts, leading to notable shape distinctions. Nonetheless, the mention of “pop-out” occurring in the letter category has been removed, as the necessity of introducing pop-out in our gaze-contingent display, which forces participants to visual search in a serial manner, was deemed unnecessary.

6. The purpose of analyzing response time in our study was to investigate the impact of delay on participants' behavior, paralleling our analysis of eye movements. By comparing target and no-target conditions, we also assessed whether participants anticipated the target's location and potentially skipped trials before completing the visual search. This analysis was crucial to ensure that the eye movement data we collected accurately represented the participants’ behavior during the visual search. Although response times for no-target trials were shorter than double for target trials, the statistical significance between these conditions indicates that concerns about participants “skipping” trials are not substantial. While this does not conclusively prove that skipping never occurred, it suggests that such behavior was not a significant factor in our results.

Furthermore, considering our study's two-step response process and the absence of instructions for participants to respond rapidly, we agree that it is not appropriate to definitively label the observed increase in response times with delay, especially for no-target stimuli, as “task difficulty.” The trends observed imply a higher level of challenge in the task with delays. However, in line with your feedback, we acknowledge the difficulty in categorically defining this as task difficulty and have therefore refrained from using this term in our revised manuscript. 

7. Regarding the last comment, the section referred to in the original manuscript (lines 358-362) described the Generalized Linear Mixed Model comparing the “other” response to the “delay” response, which does not align with the content of the comment. We believe the reviewer was referring to the paragraph discussing the critical temporal delay (original lines 594-601). We understand the comment to be highlighting that the lack of statistical significance in our results does not conclusively indicate the absence of any effect, particularly at shorter delays. Accordingly, we have revised this section in the manuscript (now lines 612-622), where we acknowledge the power deficit in our analysis. Despite this limitation, we do not entirely dismiss the potential influence of shorter delays, as evidenced by the early deviation tendency observed in our data.

Notification regarding the revisions that do not belong to the comments:

1. The analysis of fixation count in three areas of interest, originally detailed in lines 401-426, has been moved to Appendix 4. This adjustment was made because the primary aim of this analysis was to assess any unexpected eye behavior, which is ancillary to our main objective of investigating the impact of delays on eye movements. We now briefly mention this analysis in the stimuli section to demonstrate that participants did not exhibit unexpected behavior in response to our stimuli (revised lines 326-329).

We would like to clarify that there was no additional external funding received for this study. We appreciate the journal's assistance in updating the online submission form to reflect this accurate funding statement.

We hope that the revisions made to our manuscript now meet the journal’s requirements. We look forward to working again with you and the reviewers to move this manuscript closer to publication in PLOS ONE.

Thank you for your consideration. We look forward to hearing from you.

Sincerely,

Junhui Kim

School of Engineering, Tokyo Institute of Technology

Room 501, 5th Floor, Ishikawadai Building No.1, 1 Chome-1 Ishikawacho, Ota City, Tokyo

145-0061, Japan

+81 80 3059 1368

Email: kim.j.ba@m.titech.ac.jp

---

## [Decision Letter · Decision Letter 1]

13 Mar 2024

PONE-D-23-32262R1Sense of agency at a temporally-delayed gaze-contingent displayPLOS ONE

Dear Dr. Kim,

Thank you for submitting your manuscript to PLOS ONE. After careful consideration, we feel that it has merit but does not fully meet PLOS ONE’s publication criteria as it currently stands. Therefore, we invite you to submit a revised version of the manuscript that addresses the points raised during the review process.

We look forward to receiving your revised manuscript.

Kind regards,

Michael B. Steinborn, PhD

Section Editor

PLOS ONE

**Additional Editor Comments:**

I have now received feedback from the two initial reviewers (R1 was positive, while R2 was initially negative), and I further sought additional opinions from another expert (R3). R1 now finds the manuscript ready for publication, while R2 has moderated his stance a bit but still has some remaining issues that need consideration in another revision. It is also noteworthy that R3’s feedback aligns well with the original points by R1 and R2, particularly with regard to the need to clarify theoretical arguments for (model-based) predictions and, accordingly, to improve the discussion. These comments should be addressed to clarify issues previously highlighted by the other reviewers. I recommend preparing a revised submission, accompanied by a detailed letter that addresses each comment point by point. Good luck with the revision.

Reviewers' comments:

Reviewer's Responses to Questions

**Comments to the Author**

1. If the authors have adequately addressed your comments raised in a previous round of review and you feel that this manuscript is now acceptable for publication, you may indicate that here to bypass the “Comments to the Author” section, enter your conflict of interest statement in the “Confidential to Editor” section, and submit your "Accept" recommendation.

Reviewer #1: All comments have been addressed

Reviewer #2: (No Response)

Reviewer #3: (No Response)

2. Is the manuscript technically sound, and do the data support the conclusions?

Reviewer #1: (No Response)

Reviewer #2: Partly

Reviewer #3: Partly

3. Has the statistical analysis been performed appropriately and rigorously? 

Reviewer #1: (No Response)

Reviewer #2: Yes

Reviewer #3: Yes

4. Have the authors made all data underlying the findings in their manuscript fully available?

Reviewer #1: (No Response)

Reviewer #2: Yes

Reviewer #3: Yes

5. Is the manuscript presented in an intelligible fashion and written in standard English?

Reviewer #1: (No Response)

Reviewer #2: Yes

Reviewer #3: No

6. Review Comments to the Author

Reviewer #1: (No Response)

Reviewer #2: I thank the authors for their efforts in revising the manuscript. The revision significantly improved the logic of the manuscript; I mean, my previous critical concern has been addressed. I now understand the authors' primary argument. The authors investigated the sense of agency in terms of the action-feedback discrepancy using the gaze-contingent display. This study may have a theoretical ground of a comparator model, but it does not directly and empirically address this model in detail. Despite this theoretical uncertainty, this study provided important findings because of the real technological situation of widespread gaze interfaces, possibly with temporal delay.

Nevertheless, I suggest further refinement of the paper. The revised manuscript separately clarified what was directly empirically addressed and what was speculative but theoretically probable. If this holds, expanding theoretical considerations in Discussion is not precluded but rather encouraged.

Moreover, I still recommend a reader-friendly narrative for several descriptions to follow the sequence of providing a more general description followed by a detailed description. It is essential to try to avoid misleading the general reader. Indeed, I gave negative feedback on the first peer review but have now altered it by looking at the current improved descriptions. See below for detailed comments.

1) It would be fruitful to have a detailed speculative discussion about the theory as far as the authors define it as speculation. I encourage the authors to provide an exhaustive, if not integrated, discussion of the following results: the authorship rating significantly decreased during the temporal discrepancy between 200-300 ms, the mode of fixation duration (one mode reflecting wait-and-read) progressively shifted between 0-500 ms, and only one strategy (pre-registering) was adopted above 500 ms. I believe that a comprehensive discussion of these results will enhance the value of the results and eliminate possible misunderstandings.

First, the authors devote much of the paper to describing the two modes of fixation duration between 200 and 500 ms temporal discrepancies. However, in the context of this study, the important point seems to be the degree to which one mode reflecting the wait-and-read strategy gradually shifted between the discrepancy of 0 to 500 ms. This is because another mode reflecting pre-registering may only reflect a task-specific search strategy that primarily occurs in a discrepancy range of 1,000 to 4,000 ms, which is clearly beyond the time range that can be ascertained only by the comparison of afferent and efferent signals (this led to my earlier objection to the suggestion of a comparator model based on the results). Is it possible to, for example, statistically reveal that the mode reflecting the wait-and-read strategy did not shift much in the discrepancy range from 0 to 100 ms, but suddenly shifted from 200 to 300 ms? That is, if you want to leave a claim about the comparator model, you may extract the trials in which the read-and-wait strategy was employed to see if such a result can be obtained (this is not mandatory, since it may require a great deal of effort to manually browse all the data). If you observe such a fact, it may be possible to argue, for example, that the sense of agency or the comparison of afferent and efferent signals (critically changing at 200-300 ms) is behind the behavioral changes in gazing found here. Whether or not an analysis is performed, we recommend at least explaining these details in the discussion.　

On the other hand, is the degree of sense of agency relevant to why wait-and-read strategy is no longer taken at the temporal discrepancy of 1,000 ms or more? It is not clear whether you interpret this phenomenon as having something to do with authorship rating. You seem to be explaining it in the different viewpoint from authorship rating, such as referring to the limitations of memory (lines 570-572 in clean copy); however, at the same time, you are basing your interpretation on prior research on sense of agency (lines 573-576).

Elaborating on these aspects may facilitate the interpretation and exclude possible misunderstandings. If these interpretations are unclear, as I did previously, it could be read as if the one mode (pre-registering strategy) in the range of 1,000-4,000 ms discrepancy, different from the two modes in other range, implies the function of the sense of agency or comparator model. As I pointed out previously, adopting the strategy (i.e., pre-registering) for a particular task in response to such a cognitively detectable, explicit temporal discrepancy can be explained without the concept of a sense of agency and, particularly, the comparator model. Conversely, discussion that emphasizes the shift of one mode reflecting wait-and-read strategy may provide a theoretical contribution to the sense of agency and the comparator model.

2) The introduction explaining the purpose of this study, although noticeably improved, still seems to need to flow better and aid the reader's understanding. Could the sentences in lines 83-100 be made to connect point-by-point with the hypotheses, if possible, while emphasizing what is directly relevant to your experimental design (i.e., the contents described in lines 124-136)?　

3) In the method description, I recommend that you first describe your intent in your study and then supplement your reasons.

As for lines 202-245, my previous suggestion was to summarize and describe the important procedures here, not to describe all the details first. I suggest that only the important details be described here. Nevertheless, I agree that the experimental procedure is rather complex and that it will not be conveyed to the reader unless the procedural details are described to some extent here.

In lines 236-243, I think, the statement should be in the order that you wanted to use peripheral blurring for several reasons, and therefore, this required reducing the problem of saccade amplitude effects, resulting in the current methodology.

Minor (formal) comments

a) Generally, in a reply letter to review comments, the authors write the reply after describing the reviewer's original comments. In some systems, the reviewers cannot see their previous comments and may have inconvenience.

b) In the results, some ambiguity remains in the description. Figures 3-5 illustrate the performance in the playback condition. In this condition, is the participant performing a visual search or not? When I saw the word "playback" alone, I imagined that there was no search. It is recommended to explain it.

c) Is it common or in accordance with PLOS guidelines to insert a limitation section after the conclusion?　Personally, I don't see these sequences very often (I often see the limitation before the conclusion).

Reviewer #3: The authors investigated the sense of agency for eye movements and its association with temporal delays. They administered a gaze contingent window and delayed its movement by 0-4000ms. They found that temporal delay lead to decreased sense of agency, which is well in line with previous research. I find the paper interesting and appreciated the connections to the applied research field of HCI. However, I think that the manuscript should be restructured in some crucial sections to increase clarity. Further, I find the discussion lacking and somewhat missing the point of sense of agency for eye movements, as the authors tend to emphasize other aspects of eye movements and action control, rather than agency. I think the manuscript would greatly benefit by a thorough revision of the discussion. See below for detailed comments:

1. The title and introduction of your manuscript suggest a primary focus on the sense of agency in relation to eye movements. This sets up an expectation for the reader that the core findings and discussions will revolve significantly around this theme. However, upon reviewing the discussion section, I've noticed that the main results, specifically the agency ratings affected by the manipulations of delay and gaze window, are addressed only comparably briefly (l. 493-537 and l.612-622). Instead, there appears to be a substantial emphasis on what seems to be exploratory findings related to the suspected action strategies of the subjects (l. 537-611).

These exploratory findings, while intriguing, were not well grounded in the theoretical framework laid out in the initial sections of your paper. Given their speculative nature and the relatively small sample size of your study, these findings require cautious interpretation. Yet, they seem to be presented with considerable emphasis, overshadowing the more critical discussions surrounding the sense of agency, which is purportedly the central theme of your study. Furthermore, these strategies are not discussed in relation to the sense of agency but stand somewhat on their own.

I recommend revising the discussion section to rectify this imbalance. A more thorough exploration of the sense of agency, particularly how it is influenced by your experimental manipulations and how this fits to aspects from your introduction (such as the comparator model, etc.), would align better with the expectations set by your paper's title and introduction. While the exploratory findings regarding action strategies can still be included, they should be contextualized appropriately within the broader narrative of your study, and their speculative nature should be clearly articulated. This approach would not only strengthen the coherence of your manuscript but also ensure that the primary focus remains on the sense of agency, as initially suggested.

2. In your manuscript, you mention at several junctures (notably lines 101-107 and 146-148) that your work is the first to systematically look into the association between sense of agency for eye movements and temporal delays. Further, you report a notable scarcity of research connecting eye movements and sense of agency, referencing only the study by Grgič et al. While I agree that the sense of agency for the oculomotor action domain may not be as extensively explored as others, such as in the manual action domain, I'd like to point out that there exists a broader range of studies than suggested. In particular, there have been investigations into the sense of agency in gaze-contingent paradigms, including systematic analyses of temporal discrepancies or delays. Here are some references that may enrich your discussion:

• https://doi.org/10.1007/s12193-018-0286-y

• https://doi.org/10.1016/j.actpsy.2023.104121

• https://doi.org/10.3758/s13428-019-01299-x

I recommend refining your statements about the uniqueness of your research to reflect these contributions. It would also strengthen your paper to integrate these studies into your introduction and consider them when discussing your results.

For full transparency, I want to clarify that I am one of the co-authors of the paper titled "Eye did this! Sense of agency in eye movements". My suggestion to include this and the other studies is not meant to enforce citation of my own work. Instead, I believe they offer relevant insights that could enhance the depth and context of your manuscript.

3. In your manuscript, I noticed that the exact task assigned to participants was not clearly detailed, specifically regarding the concrete instructions they were given. Understanding the precise instructions is crucial for interpreting the participants' responses and the overall outcomes of your study. For instance, in "Questionnaire 1," participants were asked to rate whether the window shift could have been caused by some "other" person or entity. This prompts me to wonder about the context in which they might consider someone else as the author of the action.

In the study by Farrer et al., which you cited, it is clear that participants were informed that the movement of the virtual hand could also be caused by the experimenter. This setup provides a logical basis for participants to potentially attribute the movement to another person. However, it remains unclear whether a similar context was established in your study. Did your participants receive instructions that led them to believe that the movements of the gaze window they observed could be externally induced, perhaps by the experimenter or another entity? Clarifying this aspect would greatly enhance the understanding of why participants might attribute actions to external sources, aligning or contrasting with the precedent set by Farrer et al. Please provide detailed information on the instructions given to participants to help clarify this issue. The results should also be discussed with this in mind.

4. The Methods section of your manuscript appears to lack clear organization, which could hinder the reader's understanding of your study's design and execution. To improve clarity and coherence, I recommend restructuring this section into the following subheadings:

1. Participants

2. Apparatus and Stimuli

3. Procedure and Task: I suggest consolidating the content from lines 202-245 and 331-341 under this heading. Within this subsection, I would suggest beginning by explaining the task that participants were required to perform. Then I would follow this with a description of the different visibility conditions implemented in the study. Next, I would outline the agency questions posed to participants, ensuring to clarify the context and purpose of each. Finally, I would detail the block structure of the experiments, including how tasks and conditions were ordered and any variations between different conditions.

Furthermore, I note a lack of a dedicated section for statistical analysis. Such a section is crucial for understanding how the data were treated and interpreted. Please create a separate section for Statistical Analysis, incorporating the relevant details currently found in lines 366-371, 390-395, and 423-425. In this new section, describe the analysis methods used, the factors considered, alpha-levels set for significance, any post-hoc tests applied, etc.

5. I think that your manuscript would also benefit from some clarifications:

• You refer to misattribution of agency in several sections, e.g., “First, participants will not misattribute with temporal delays in eye-gaze HCI, as seen in the previous findings [22].” (l.126-127) or in l. 495. This phrasing is somewhat difficult to comprehend, please clarify what is meant by “misattribute”

• In a similar manner, you write “Moreover, it has been proposed that the two choices of 'self' and 'other' do not fully cover scenarios where participants perceive their actions as contributing to an outcome, but not completely or directly.” (l. 155 - 157). Please clarify what is meant by “perceiving their actions as contributing to an outcome, but not completely or directly”. You could also refer to this explanation when you discuss the task instruction (see reviewer comment #3)

• I struggle to fully understand the two strategies (wait-and-read vs. pre-registering) in section l. 538-552. Could you please further clarify these strategies and how they differed from each other?

Minor Concerns:

6. You state, that sense of agency “cannot be adequately represented by a continuous degree rating.” (l. 155). I think that the reasoning behind this statement should be better explained and carefully debated. For me, it makes sense that the adapted Farrer questionnaire still leaves an option open to differentiate between “self”, “delayed”, and “other”. However, you should briefly explain what you consider to be the specific problems (and benefits?) of a continuous scale in contrast to the Farrer-questionnaire. A more differentiated consideration seems particularly useful to me here because you yourselves use a continuous scale to measure sense of agency in the later text (l.164-174).

7. I would encourage you to give some form of sample size justification even if only acknowledging the absence of any justification. A good guide for this is provided by the paper by Lakens, 2021: https://www.semanticscholar.org/paper/Sample-Size-Justification-Lakens/c591254d1746341feae3eb6d217c970ff4bc9a87

8. The section l. 452-456 should be moved to the Apparatus section.

9. In your Results section, you refer to your questionnaires as “First Questionnaire” and “Second Questionnaire”. I would prefer a more meaningful naming, for example “categorical” vs. “continuous” questionnaire, etc.

10. I found the text in l. 380 – 384 to be redundant to the figure description in l. 386-388. I think that the figure description is sufficient and the text in l. 380-384 could be deleted or shortened.

11. You give information about the mean differences between conditions for your post-hoc tests in some sections (e.g., l. 358-360) but in other sections you just report p-values without condition means (l. 363-365, l. 442-444). Please also report the means in these sections so that the reader can understand in which direction the effect is pointed and how large the effect might be.

12. Last, I would encourage you to report some measure of effect size for your ANOVAs and mixed models. This might be especially helpful in regard to the small sample size. You state in the Limitations section that your study might have been underpowered, but this can only be determined if the effect sizes are estimated.

I want to thank the authors and the editor for giving me the opportunity to review this paper. I hope that my feedback is helpful.

7. PLOS authors have the option to publish the peer review history of their article (what does this mean?). If published, this will include your full peer review and any attached files.

Reviewer #1: No

Reviewer #2: No

Reviewer #3: **Yes: **Julian Gutzeit

---

## [Author Response · Author response to Decision Letter 1]

4 Jul 2024

5-July-2024

Michael B. Steinborn 

Section Editor

PLOS ONE

Dear Editor:

We wish to re-submit the manuscript titled “Sense of agency at a temporally-delayed gaze-contingent display.” The manuscript ID is PONE-D-23-32262R1.

We would like to express our sincere gratitude to the editor and reviewers for their valuable time and effort in reviewing our paper and providing constructive feedback that has significantly contributed to the improvement of our manuscript. We are also grateful for the addition of a new reviewer, whose insights have added immense value to our work. Additionally, we appreciate the multiple extensions granted, which have allowed us to further refine our research. We have made the necessary changes according to the feedback provided. We believe that these revisions have enhanced the quality of our work. Detailed responses addressing the concerns raised by the editor and reviewers can be found below.

We appreciate reviewer 2's guidance that previous comments might not be visible to reviewers in the system. To address this, we have copied the full text of the reviewers' original comments and provided point-by-point responses.

Response to the comments from Reviewer 2:

・Comment 1:

It would be fruitful to have a detailed speculative discussion about the theory as far as the authors define it as speculation. I encourage the authors to provide an exhaustive, if not integrated, discussion of the following results: the authorship rating significantly decreased during the temporal discrepancy between 200-300 ms, the mode of fixation duration (one mode reflecting wait-and-read) progressively shifted between 0-500 ms, and only one strategy (pre-registering) was adopted above 500 ms. I believe that a comprehensive discussion of these results will enhance the value of the results and eliminate possible misunderstandings.

First, the authors devote much of the paper to describing the two modes of fixation duration between 200 and 500 ms temporal discrepancies. However, in the context of this study, the important point seems to be the degree to which one mode reflecting the wait-and-read strategy gradually shifted between the discrepancy of 0 to 500 ms. This is because another mode reflecting pre-registering may only reflect a task-specific search strategy that primarily occurs in a discrepancy range of 1,000 to 4,000 ms, which is clearly beyond the time range that can be ascertained only by the comparison of afferent and efferent signals (this led to my earlier objection to the suggestion of a comparator model based on the results). Is it possible to, for example, statistically reveal that the mode reflecting the wait-and-read strategy did not shift much in the discrepancy range from 0 to 100 ms, but suddenly shifted from 200 to 300 ms? That is, if you want to leave a claim about the comparator model, you may extract the trials in which the read-and-wait strategy was employed to see if such a result can be obtained (this is not mandatory, since it may require a great deal of effort to manually browse all the data). If you observe such a fact, it may be possible to argue, for example, that the sense of agency or the comparison of afferent and efferent signals (critically changing at 200-300 ms) is behind the behavioral changes in gazing found here. Whether or not an analysis is performed, we recommend at least explaining these details in the discussion.　

On the other hand, is the degree of sense of agency relevant to why the wait-and-read strategy is no longer taken at the temporal discrepancy of 1,000 ms or more? It is not clear whether you interpret this phenomenon as having something to do with authorship rating. You seem to be explaining it from the different viewpoint from authorship rating, such as referring to the limitations of memory (lines 570-572 in clean copy); however, at the same time, you are basing your interpretation on prior research on sense of agency (lines 573-576).

Elaborating on these aspects may facilitate the interpretation and exclude possible misunderstandings. If these interpretations are unclear, as I did previously, it could be read as if the one mode (pre-registering strategy) in the range of 1,000-4,000 ms discrepancy, different from the two modes in other range, implies the function of the sense of agency or comparator model. As I pointed out previously, adopting the strategy (i.e., pre-registering) for a particular task in response to such a cognitively detectable, explicit temporal discrepancy can be explained without the concept of a sense of agency and, particularly, the comparator model. Conversely, discussion that emphasizes the shift of one mode reflecting wait-and-read strategy may provide a theoretical contribution to the sense of agency and the comparator model.

・Response 1:

The reviewer suggests an exhaustive discussion of results showing authorship rating decreases, fixation duration mode shifts, and the adoption of a pre-registering strategy. The reviewer questions the relevance of a sense of agency in strategy changes at different delays and suggests explaining these phenomena clearly in the discussion to avoid misunderstandings. The reviewer also proposes potentially revealing statistical shifts in fixation strategies to support claims about the comparator model and sense of agency.

1. The reviewer suggested focusing on how the wait-and-read strategy shifts between 0 to 500 ms rather than the pre-registering strategy, which might be task-specific and occur beyond 1,000 ms.

2. The reviewer recommends statistically revealing that the wait-and-read strategy does not shift much from 0 to 100 ms but shifts significantly from 200 to 300 ms to support claims about the comparator model.

3. The reviewer questioned the relevance of the sense of agency to the wait-and-read strategy's discontinuation at delays over 1,000 ms, noting an unclear interpretation.

4. The reviewer suggested elaborating on these aspects to avoid misunderstandings, indicating that a shift in strategy without the concept of the sense of agency can be explained by cognitive and memory constraints.

We deleted the interpretation of strategies change with the framework of sense of agency (lines 573-576 in the original manuscript). This explanation was the confounding factor caused by our aim to discuss how the two strategies co-existed. We had to clearly separate the explanation of memory limitation (lines 569-573 in the original) for strategy change and sense of agency (lines 573-576 in the original) for two co-existing strategies. Nonetheless, the original manuscript was a confounding discussion that the sense of agency can be the explanation of how strategy changes at the delays of 1000 ms. The original lines 573-576 were intended to argue that participants might tend to apply different strategies to different conditions, similar to Wen's (2019) assertion that participants tend to give different rating scores for different conditions because the reported sense of agency was biased by the graded delay conditions (https://doi.org/10.1016/j.concog.2019.05.007). With the suggested analysis, we replaced the discussion of co-existing two strategies in the 200-500 ms delay conditions (lines 568-569 in original), with the shifting mode from 0 to 500 ms delay conditions (line 858 in revision). We consider the suggested analysis as a more suitable approach in discussing the results of fixation duration distribution in the 0-500 ms delays, with a sense of agency. 

We appreciate the reviewer’s analysis suggestion that we have not approached. As the reviewer commented, we chose not to extract the interesting trial of “wait-and-read” as it requires extensive time. Rather, we chose to analyze the characteristics of the distribution (Fig 6, Fig 7) we plotted from the eye movements data. For the “wait-and-read”, we calculated and analyzed the x-axis location of the shifting mode in the fixation duration distribution at the delay range from 0 to 500 ms. Also, we analyzed the response proportion of “self”. We chose a categorical response instead of the rating scale. This was because we considered categorical “self” responses revealed a more radical shift in the delay range from 0 to 500 ms, thus providing more power in statistical analysis. By this decision, the discussion topic related to the rating score was limited to 1) the comparison with prior studies, and 2) the tendency inspected by the graph, which was the decrease with a concave upward pattern (discussed in lines 795-807 in revision). While investigating the rating scale in this analysis might provide rich discussion, we believe the rating scale has fulfilled its role (addressing the second hypothesis) adequately. We described why the rating score was not considered with this analysis in lines 470-474 in the revision. Further, we analyzed the cumulative distribution of saccades at 1.5 degrees to discuss how the variation with a delay of 0 to 500 ms could be related to the sense of agency. We plot each value with a function of delay with the range from 0 to 500 ms (Fig 8 A-C). Further, we calculated the difference between the two delays with 100 ms intervals (e.g., change in 100 – 0, 200- 100, 300- 200, 400-300, 500-400 ms), which we called contrast (Fig 8 D-F). The figure caption can be found in lines 644-652 in the revision. Statistical analyses were done on both value and contrast.

Throughout the manuscript, we refer to the suggested analysis as “contrast analysis”. In the delay range of 0-500 ms, we analyzed the following three: 1) the proportion of “self” response, 2) the location of shifting mode in the fixation duration distribution, and 3) the cumulative distribution of saccades at 1.5 degrees. The method of how we conducted the analysis, including the method by which we examined the x-axis location of the shifting mode, was described in the newly created subsection of Analyses in the Method section (lines 475-500 in revision). Results were described in lines 636-682 in the revision. Discussions were described throughout different sections. Discussion for “self” response was described in lines 752-767, fixation duration shifting mode was discussed in lines 872-888, saccade under 1.5 degree was discussed in lines 910-919, in revision.

In the revised discussion regarding the fixation duration, we first described the two strategies (lines 816-837 in revision), how two strategies were identified (by observing experiment and recorded data, lines 838-845 in revision), and how the strategies explain the distribution function (lines 845-855 in revision). Then, in the next paragraph (lines 856-888 in revision), we separated the discussion between strategy transition (at 1000 ms delay) and the shifting mode. We discussed that the strategy transition could be explained by the participants’ memory limitation and would not related to the sense of agency (lines 858-872 in revision). Then, we discussed how the shifting mode would relate to the sense of agency (lines 872-888 in revision). 

To provide a reader-friendly narrative for this discussion, we articulated in the first paragraph of the subsection Behaviors in Discussion section, that we discuss participants’ task strategies and how task strategies might related to the sense of agency (lines 809-815 in revision). With this revision, we further clarified our approach to discussing the sense of agency.

We did not introduce this contrast analysis in the introduction section as the analysis. We considered the contrast analysis to rather post-hoc analysis. It aligns well with our original hypotheses. Analysis of “Self” allowed us to argue the comparator model would be related to the decrease of “self” response, thereby providing insights to our first hypothesis that participants did not judge “other” but ‘delayed but self’ (“delay” response), and the “delayed but self” judgment might be explained with the comparator model. Contrast analysis with eye movements serves our third hypothesis that the changes in 0-300 ms can be explained by the action refinement in the comparator model.

Contrast analysis revealed that significant changes occurred from 0 to 200 ms in “self” response, and 100 to 200 ms in fixation duration shifting mode location. We think these results are well aligned with the comparator model as 200-300 was frequently reported to the delay where body agency (feeling of agency) is maintained. The feeling of agency is the term coined by Synofzik, 2008 (https://doi.org/10.1016/j.concog.2007.03.010) whereas the body agency is the term provided by Wen, 2019 (https://doi.org/10.1016/j.concog.2019.05.007). Both terms share the same definition of authorship feeling towards own action.

・Comment 2:

The introduction explaining the purpose of this study, although noticeably improved, still seems to need to flow better and aid the reader's understanding. Could the sentences in lines 83-100 be made to connect point-by-point with the hypotheses, if possible, while emphasizing what is directly relevant to your experimental design (i.e., the contents described in lines 124-136)?　

・Response 2:

To flow better, we deleted the last two sentences from the original paragraph (lines 97-100 in the original). The two sentences introduced that the two factors, arousal, and performance, impacted the sense of agency. The two factors were irrelevant to the current independent variable, action-effect temporal discrepancy (delay). The exact two studies, at the same time, employed the delay as additional independent variables. Aside from the last two sentences, the two studies were still referred to throughout this manuscript when we compare our results of the delay’s effect on the sense of agency.

As recommended by the reviewer, we added three descriptions. Lines 94-96, 101-106, and 112-116. The first and second descriptions connect to our first hypothesis, and the third description connects to our second hypothesis. 

・Comment 3:

In the method description, I recommend that you first describe your intent in your study and then supplement your reasons.

As for lines 202-245, my previous suggestion was to summarize and describe the important procedures here, not to describe all the details first. I suggest that only the important details be described here. Nevertheless, I agree that the experimental procedure is rather complex and that it will not be conveyed to the reader unless the procedural details are described to some extent here.

In lines 236-243, I think, the statement should be in the order that you wanted to use peripheral blurring for several reasons, and therefore, this required reducing the problem of saccade amplitude effects, resulting in the current methodology.

・Response 3:

We agree with Reviewer 2’s advice to prioritize important information. At the same time, we also agree with Reviewer 3’s Comment 4 to organize the method section by separating it into distinct subsections. For lines 236-243 in the original manuscript, we have introduced the rationale for using peripheral blurring at the beginning of the Stimuli section to ensure readability (lines 282-299 in revision). The rationale for using blurring is introduced first: it was implemented to enforce serial search and effectively disrupt behavior (lines 282-285 in revision). We addressed the concern of decreasing saccade amplitudes by peripheral blurring (lines 285-292 in revision). Then, we described the current design (lines 293-299 in revision).

We have separated the visibility condition into its own paragraph for clarity (lines 311-324 in revision). To help readers’ understanding the sentence describing the visibility conditions (originally in lines 227-228) was placed at the first of that paragraph (lines 311-312 in revision).

In the Stimuli subsection, we divided the details on how the Chinese characters were pooled and how the target characters were placed into two separate paragraphs and placed them at the end of the section (lines 336-355 in revision). With this revision, we believe the important information regarding gaze-contingent windows and stimuli was placed first in the subsection of Stimuli. Now the Stimuli subsection has the flow of the following: rationale and description of our stimuli design (lines 282-299 in revision

---

## [Decision Letter · Decision Letter 2]

24 Jul 2024

PONE-D-23-32262R2Sense of agency at a temporally-delayed gaze-contingent displayPLOS ONE

Dear Dr. Kim,

Thank you for submitting your manuscript to PLOS ONE. After careful consideration, we feel that it has merit but does not fully meet PLOS ONE’s publication criteria as it currently stands. Therefore, we invite you to submit a revised version of the manuscript that addresses the points raised during the review process. **Editor comments**. I have now received all reviews and, based on their opinions, a decision can be made. All reviewers were constructive and engaged in an excellent dialectical discourse, which is outstanding and aligns with the scientific ideal we aspire to. At present, all three reviewers are generally satisfied and find the manuscript in proper shape, although they have some remaining comments. R3, however, has additional issues that must be resolved before publication, but these will not take much time to address in the final preparation of the manuscript. We are now close to the final stage, and I ask you to revise the manuscript in a final revision, which I believe will likely be the last round. Please provide a point-by-point response to the comments.

We look forward to receiving your revised manuscript.

Kind regards,

Michael B. Steinborn, PhD

Section Editor

PLOS ONE

Journal Requirements:

**Additional Editor Comments:**

In the following, I provide comments on various issues I find relevant for the revision. Additionally, I will address and discuss comments from all reviewers to clarify their potential implications and how to handle them in the final manuscript revision. I want to emphasise that my comments are intended to improve the manuscript, not to criticise your work. I do not expect you to agree with all of my points; in other words, I do not claim that my opinion is always correct. I see myself more as a moderator, facilitating the interaction between authors and reviewers to enhance the final quality of the manuscript. I have no claims to absolute truth or insistence on my views, just to clarify.

(-1-) models of agency and temporal delay

The authors argue that participants did not misattribute authorship with temporal delays, which in their interpretation, strongly supports the hypothesis that such delays are not strong enough to cause misattribution. Based on the comments from R1, R2, and R3, I suggest presenting a more elaborate interpretation in the final revision of the manuscript. Here are some ideas, mainly derived from the reviewers' comments: Why and when do we perceive a sense of agency during our actions? And when are these perceptions strong enough to change how we view an action or our role in it? Let me explain, bearing in mind my perspective as a layman reader of your manuscript. Whether we perceive a sense of agency during an ongoing action sequence or a performed action depends on the temporal proximity between our (a) intention formation, (b) action initiation, and (c) the consequence of the action (i.e., the outcome of the action) (supporting refs: doi:10.1007/s00221-020-05861-4). This is clear in everyday situations. For example, when an individual presses a light switch, they observe their finger move, feel the switch haptic feedback, and see the light turn on almost instantaneously. This immediate temporal coupling between the action and the sensory feedback leads us to "know" we are the cause and intended agents of that action.

We can distinguish between two categorical kinds of agency: one refers to the ongoing process (perceptual regulation), and the other refers to the outcome of the intended action (outcome control). Both contribute to the feeling of "authorship" of the action, depending on the situational model. This means we can perceive something as "caused by us" even if it is not immediately contingent, because we often rely on internal representations of expected temporal delays. For instance, when pressing a button on a coffee machine, there is a "known" delay of about 1 minute before the coffee is dispensed. Despite this delay, individuals still perceive themselves as the "author" of the action, as their internal model includes the temporal parameters of the situation. This internal representation allows for a sense of agency even when the outcome is not immediate, as long as the delay is within the expected timeframe of the individual mental model. In the context of the current study, the temporal delays introduced were likely ot sufficient to disrupt the internal models or the immediate temporal contingencies necessary for the sense of agency, yet alternatively, the implanted mental model (delivered by the paradigm) was not clear enough. Why is this the case? It is difficult to say definitively, as I am not an expert, but fundamentally, participants retained their sense of authorship despite the delays, likely because these delays were within the bounds of their internal representations of how long actions and outcomes should be linked. This highlights an important aspect that needs further discussion: the concept of "situational model ambiguity". Lewin (1943) referred to this as "strong vs. weak" model situations. This concept should be elaborated upon in the discussion section to provide a deeper understanding of the findings.

(-2-) wait-and-read vs. pre-registering

Two main strategies were identified: "wait-and-read" and "pre-registering," demonstrating how participants adapted to varying delays. Based on the arguments of R2 and R3, it seems that we are observing aspects of perceptual control (which the authors conceptualise more as preregistered online control) and model-based control (outcome control). I suggest elaborating more on the critical experimental factors that comprise the design and collectively conceptualise the representable model situation, rather than focusing on technical details.

R2 argues that although spatial attention is acknowledged, it is only briefly mentioned by name. R2 (but also R3 in some sense) ague that the discussion should emphasise this more by elaborating more deeply on temporal attention, as it aligns more closely with the present main hypothesis. For example, to make that point clear, R2 notes that oculomotor events, such as pupil diameter regulation and gaze position stability, are closely linked to the timing of target appearance, and this can be discussed in two ways: first, it concerns the specific model situation being addressed. The authors discuss the impact of delays in contingent displays and have implications for designing delay-tolerant gaze-contingent systems, indicating that they already have specific model situations in mind. However, these are not fully articulated. Second, there is a fundamental aspect related to the basic structural conditions of temporal preparation. This aspect should be prioritised and more thoroughly examined in the discussion to provide a comprehensive understanding of the findings. Suggested references supporting these arguments further are here:

Cao, L. et al. (2020). Action force modulates action binding: Evidence for a multisensory-integration explanation. Experimental Brain Research. doi:10.1007/s00221-020-05861-4

Logan, G. D., & Crump, M. J. C. (2010). Cognitive illusions of authorship reveal hierarchical error detection in skilled typists. Science, 330(6004), 683-686. doi:10.1126/science.1190483

Powers, W. T. (1973). Feedback: beyond behaviorism. Science, 179(4071), 351-356. doi:10.1126/science.179.4071.351

Yamashita, J. et al. (2022). Pupillary fluctuation amplitude preceding target presentation is linked to the variable foreperiod effect on reaction time in Psychomotor Vigilance Tasks. Plos One, 17(10), e0276205. doi:10.1371/journal.pone.0276205

(-3-) novelty

Both R2 and R3 express concerns about the authors presenting their findings as absolutely novel or unprecedented, requesting corrections. R3 questions whether the results confirm previous knowledge or offer novel insights capable of reshaping the theoretical landscape, arguing that it is unclear what new contributions this study makes. Meanwhile, R2 emphasizes the importance of understanding the mechanics underlying perceptual agency and authorship. While I fully agree with these points, I want to defend the authors by clarifying that the aim of cognitive psychological research should be to systematically delineate mechanisms, such as how people represent actions and their consequences, rather than constantly aiming to surprise. Otherwise, we risk falling into the trap of sensationalist reporting that prioritises eye-catching headlines over factual accuracy and integrity. "Novelty" should not be misconstrued as "discovery," especially in a field prone to false-positive results. Therefore, a lack of novelty is not problematic simply because an effect or correlation has been reported previously, as it is more crucial, at least in my view, to recognise that many findings in a field are often inconsistent, and demonstrating systematic experimentation is therefore valuable. I suggest the authors therefore to elaborate and outline (at the end of the discussion) how an improved (modified or optimal) experimental approach would look like and how this can be implemented in future research on this subject. 

Reviewers' comments:

Reviewer's Responses to Questions

**Comments to the Author**

1. If the authors have adequately addressed your comments raised in a previous round of review and you feel that this manuscript is now acceptable for publication, you may indicate that here to bypass the “Comments to the Author” section, enter your conflict of interest statement in the “Confidential to Editor” section, and submit your "Accept" recommendation.

Reviewer #2: (No Response)

Reviewer #3: All comments have been addressed

2. Is the manuscript technically sound, and do the data support the conclusions?

Reviewer #2: Partly

Reviewer #3: Yes

3. Has the statistical analysis been performed appropriately and rigorously? 

Reviewer #2: Yes

Reviewer #3: Yes

4. Have the authors made all data underlying the findings in their manuscript fully available?

Reviewer #2: Yes

Reviewer #3: Yes

5. Is the manuscript presented in an intelligible fashion and written in standard English?

Reviewer #2: Yes

Reviewer #3: Yes

6. Review Comments to the Author

Reviewer #2: Most of the comments previously noted have been addressed appropriately. One final point still needs to be addressed, but overall, I believe the manuscript is now in an acceptable condition. We thank the authors for their efforts to improve the manuscript.

In lines 889-919 (clean copy), I think the authors should discuss oculomotor stability in relation to temporal attention. Although I do not deny the involvement of spatial attention, I think a discussion related to temporal attention, which is more relevant to the main hypothesis of the manuscript, is warranted. It has been noted that when a target appears at a certain time, various oculomotor events tend to stabilize as it approaches the expected time of target presentation, perhaps to get clearer image input at that time. For example, the pupil diameter, which plays a role in regulating the intensity of the image entering the eye, is stable at the time when a potential target is likely to appear (https://doi.org/10.1371/journal.pone.0276205). Also, changes in pupil diameter may lead to fine-scale variations in gaze position as measured by the eye tracker (https://doi.org/10.1371/journal.pone.0111197). It has also been reported that the gaze position becomes stable at the time when the target is about to appear (https://doi.org/10.1016/j.neuroimage.2018.09.026;
https://doi.org/10.1167/jov.23.14.1). I recommend primarily discussing these previously found phenomena in relation to the correlation between the display lag and fine-scale variation in gaze position.

Minor remarks:

1) Recently, an article has been published by the same author dealing with a similar topic (https://doi.org/10.3389/fpsyg.2024.1364076). It does not seem to deal with the exact same hypothesis, and there may not be any problem in terms of publication ethics. Nonetheless, it may be safer to cite the article. Also, please make sure that you are not using the same text or images.

2) Because the analysis in the manuscript is voluminous, I suggest giving a brief summary of the specific hypotheses at the end of the introduction or at the beginning of the methods (if it doesn't interfere with the opinions of other reviewers). Here, you may use a shortened version of sentences in lines 685-691 as a summary.

3) When citing reference [35] in line 125, the full name is listed for some reason. Please check the citation format.

Reviewer #3: Thank you for your detailed response to my comments on the manuscript titled “Sense of agency at a temporally-delayed gaze-contingent display”. I appreciate the efforts made to address the concerns and suggestions I raised in my initial review. I think that the quality of the manuscript has increased substantially. The structure is very clear and it is easy to follow the reasoning of the authors. Moreover, the discussion is now much better balanced and focused on sense of agency. I only have some minor comments:

1) I appreciate the new, differentiated contextualization of your study in light of previous research examining the sense of agency for eye movements and temporal delay, avoiding the claim of being the first to investigate this association. In the revised version, the novelty of your approach—namely the continuous eye movements compared to single saccades in previous research—is much better highlighted. However, the phrasing in the abstract, “This study is the first attempt to systematically examine the relation between temporal discrepancy and eye movements” (lines 28-29), is still somewhat misleading or a bold claim. I would appreciate it if you could tone down this statement and briefly describe the uniqueness of your approach more clearly in the abstract, as you did in the manuscript.

2) It is unclear how the 60 sessions were distributed over the days. You wrote: “Each participant required over two months to complete 60 sessions, making it challenging to recruit additional participants” (lines 260-262). Does this mean that each session was held on a separate day, or were multiple sessions conducted in a single day? Please clarify this briefly in the Task and Procedure Section.

3) Typo in l. 474: “200-30 ms”, should be “200-300 ms”, right?

4) In Figure 3, it is difficult to distinguish between the three categories (self, delayed, and other). The plot might be easier to interpret if the positions of the data points were dodged. This issue is particularly pronounced for the points in the playback condition. However, dodging the points might lead to clutter and reduce readability. I encourage the authors to try a dodged version to determine if it improves or worsens the plot's readability. The same suggestion applies to Figures 4 and 5.

I look forward to seeing the final version of the manuscript.

Thank you for your hard work and attention to detail!

7. PLOS authors have the option to publish the peer review history of their article (what does this mean?). If published, this will include your full peer review and any attached files.

Reviewer #2: No

Reviewer #3: **Yes: **Julian Gutzeit

---

## [Author Response · Author response to Decision Letter 2]

21 Aug 2024

22-August-2024

Michael B. Steinborn 

Section Editor

PLOS ONE

Dear Editor:

We wish to re-submit the manuscript titled “Sense of agency at a temporally-delayed gaze-contingent display.” The manuscript ID is PONE-D-23-32262R2.

We sincerely appreciate the editor and reviewers for their time and effort in reviewing our paper and providing constructive feedback. Their insights have been instrumental in improving our manuscript. We are also thankful for the comments from the editor, whose perspectives have significantly enriched our work. We have implemented the necessary changes based on the feedback received. We believe these revisions have enhanced the quality of our work. Detailed responses to the concerns raised by the editor and reviewers are provided below.

Response to the comments from Editor:

・Editor, Comment 1: models of agency and temporal delay

The authors argue that participants did not misattribute authorship with temporal delays, which in their interpretation, strongly supports the hypothesis that such delays are not strong enough to cause misattribution. Based on the comments from R1, R2, and R3, I suggest presenting a more elaborate interpretation in the final revision of the manuscript. Here are some ideas, mainly derived from the reviewers' comments: Why and when do we perceive a sense of agency during our actions? And when are these perceptions strong enough to change how we view an action or our role in it? Let me explain, bearing in mind my perspective as a layman reader of your manuscript. Whether we perceive a sense of agency during an ongoing action sequence or a performed action depends on the temporal proximity between our (a) intention formation, (b) action initiation, and (c) the consequence of the action (i.e., the outcome of the action) (supporting refs: doi:10.1007/s00221-020-05861-4). This is clear in everyday situations. For example, when an individual presses a light switch, they observe their finger move, feel the switch's haptic feedback, and see the light turn on almost instantaneously. This immediate temporal coupling between the action and the sensory feedback leads us to "know" we are the cause and intended agents of that action.

We can distinguish between two categorical kinds of agency: one refers to the ongoing process (perceptual regulation), and the other refers to the outcome of the intended action (outcome control). Both contribute to the feeling of "authorship" of the action, depending on the situational model. This means we can perceive something as "caused by us" even if it is not immediately contingent because we often rely on internal representations of expected temporal delays. For instance, when pressing a button on a coffee machine, there is a "known" delay of about 1 minute before the coffee is dispensed. Despite this delay, individuals still perceive themselves as the "author" of the action, as their internal model includes the temporal parameters of the situation. This internal representation allows for a sense of agency even when the outcome is not immediate, as long as the delay is within the expected timeframe of the individual mental model. In the context of the current study, the temporal delays introduced were likely not sufficient to disrupt the internal models or the immediate temporal contingencies necessary for the sense of agency, yet alternatively, the implanted mental model (delivered by the paradigm) was not clear enough. Why is this the case? It is difficult to say definitively, as I am not an expert, but fundamentally, participants retained their sense of authorship despite the delays, likely because these delays were within the bounds of their internal representations of how long actions and outcomes should be linked. This highlights an important aspect that needs further discussion: the concept of "situational model ambiguity". Lewin (1943) referred to this as a "strong vs. weak" model situation. This concept should be elaborated upon in the discussion section to provide a deeper understanding of the findings.

・Editor, Comment 2: wait-and-read vs. pre-registering

Two main strategies were identified: "wait-and-read" and "pre-registering," demonstrating how participants adapted to varying delays. Based on the arguments of R2 and R3, it seems that we are observing aspects of perceptual control (which the authors conceptualize more as preregistered online control) and model-based control (outcome control). I suggest elaborating more on the critical experimental factors that comprise the design and collectively conceptualizing the representable model situation, rather than focusing on technical details.

R2 argues that although spatial attention is acknowledged, it is only briefly mentioned by name. R2 (but also R3 in some sense) argues that the discussion should emphasize this more by elaborating more deeply on temporal attention, as it aligns more closely with the present main hypothesis. For example, to make that point clear, R2 notes that oculomotor events, such as pupil diameter regulation and gaze position stability, are closely linked to the timing of target appearance, and this can be discussed in two ways: first, it concerns the specific model situation being addressed. The authors discuss the impact of delays in contingent displays and have implications for designing delay-tolerant gaze-contingent systems, indicating that they already have specific model situations in mind. However, these are not fully articulated. Second, there is a fundamental aspect related to the basic structural conditions of temporal preparation. This aspect should be prioritized and more thoroughly examined in the discussion to provide a comprehensive understanding of the findings. Suggested references supporting these arguments further are here:

Cao, L. et al. (2020). Action force modulates action binding: Evidence for a multisensory-integration explanation. Experimental Brain Research. doi:10.1007/s00221-020-05861-4

Logan, G. D., & Crump, M. J. C. (2010). Cognitive illusions of authorship reveal hierarchical error detection in skilled typists. Science, 330(6004), 683-686. doi:10.1126/science.1190483

Powers, W. T. (1973). Feedback: beyond behaviorism. Science, 179(4071), 351-356. doi:10.1126/science.179.4071.351

Yamashita, J. et al. (2022). Pupillary fluctuation amplitude preceding target presentation is linked to the variable foreperiod effect on reaction time in Psychomotor Vigilance Tasks. Plos One, 17(10), e0276205. doi:10.1371/journal.pone.0276205

・Editor, Comment 3: Novelty

Both R2 and R3 express concerns about the authors presenting their findings as absolutely novel or unprecedented, requesting corrections. R3 questions whether the results confirm previous knowledge or offer novel insights capable of reshaping the theoretical landscape, arguing that it is unclear what new contributions this study makes. Meanwhile, R2 emphasizes the importance of understanding the mechanics underlying perceptual agency and authorship. While I fully agree with these points, I want to defend the authors by clarifying that the aim of cognitive psychological research should be to systematically delineate mechanisms, such as how people represent actions and their consequences, rather than constantly aiming to surprise them. Otherwise, we risk falling into the trap of sensationalist reporting that prioritizes eye-catching headlines over factual accuracy and integrity. "Novelty" should not be misconstrued as "discovery," especially in a field prone to false-positive results. Therefore, a lack of novelty is not problematic simply because an effect or correlation has been reported previously, as it is more crucial, at least in my view, to recognize that many findings in a field are often inconsistent, and demonstrating systematic experimentation is therefore valuable. I suggest the authors therefore elaborate and outline (at the end of the discussion) how an improved (modified or optimal) experimental approach would look like and how this can be implemented in future research on this subject.

・Response to all the three comments:

We appreciate the thoughtful, constructive, and comprehensive feedback. We think that all three comments from the editor share similar aspects and we think that the issue presents in our unclear discussion for both short (200-300 ms) and long-lasting (several seconds) senses of agency. In the current response letter, we respond with a comprehensive explanation of the revision. We now have revised to discuss the distinction clearly and clarify that the focus of our study lies in understanding the short sense of agency, which is crucial for real-time applications such as gaze-contingent human-computer interfaces (HCI). For discussing the two types in the sense of agency, we have adopted the two-layer framework proposed by Synofzik et al. (2008) (https://doi.org/10.1016/j.concog.2007.03.010), distinguishing between the feeling of agency and the judgment of agency, and argue that our experimental design might be capable of assessing both layers. This perspective not only introduces a novelty to our study but also highlights the importance of the short sense of agency in practical applications.

The revisions include updates to the Introduction (revised lines 195-211) emphasizing the relevance of the short sense of agency for our research (corresponding to comments 1 and 2). The Discussion section now provides a clearer explanation regarding the sense of agency (revised lines 792-847), followed by a brief discussion of how our experiment relates to Synofzik’s two-layer model (revised lines 848-858) (corresponding to comment 1). Additionally, the eye movements discussion (revised lines 985-999) (corresponding to comment 2) clarifies that the “wait-and-read” strategy is linked to the short sense of agency, which is essential for real-time gaze-contingent HCI. The Future Directions section (revised lines 1091-1111) (corresponding to comments 3) and Conclusion (revised lines 1113-1127) have been revised to reflect our revisions, along with minor updates to the Abstract (revised lines 26-29).

Please note that regarding comment 2, while we discuss how our discussion of the two types in the sense of agency (revised lines 195-211) relates to the two strategies on revised lines 985-999, temporal attention was discussed somewhat independently only in the context of saccades under 1.5 degrees. This decision was made because temporal attention, while relevant, plays a distinct role from the core focus of our study on real-time gaze-contingent interactions. Prioritizing this separation allowed us to maintain the clarity of our main argument while still acknowledging the relevance of temporal attention. In the revised manuscript, we have now included a comprehensive discussion on how temporal and spatial attention may relate to the delay’s impact on saccades under 1.5 degrees (lines 1000-1028 in the revised manuscript), while also reducing the emphasis on spatial attention to better align with the relevance of temporal attention in this context. This part of our revision is also addressed in our response to Reviewer 2, Comment 1.

Further, the Introduction section was revised to flow without any misunderstanding while presenting our hypothesis more clearly. In our discussion regarding Synofzik (original lines 91-116, revised lines 180-194), we revised the sentence "However, the comparator model has been criticized [28] and described as neither necessary nor sufficient to explain the sense of agency" (lines 91-92 in original) to "Synofzik et al. [28] proposed a revised model, incorporating but expanding beyond the comparator model, which they described as incomplete to fully explain the sense of agency." (lines 182-184 in revised). We do not take a stance against the comparator model or solely favor Synofzik’s model. Our hypothesis assumes that, as previous studies suggest, efference copy exists in eye movements as well. We hypothesize that the sense of agency operates similarly in eye movements as in other actions, which justifies using both the comparator model and Synofzik’s model as explanatory frameworks. Therefore, we needed to ensure the revised sentence does not give the impression that we oppose the comparator model. For minor revision, the "helping hand" example introduced in Synofzik et al.’s discussion was removed as it was not directly related to the design of our study. Synofzik’s discussion that the questionnaire measures judgment of agency and further citation that intentional binding measures the feeling of agency now moved to the discussion section (original lines 108-112, revised lines 848-852) when we discuss how our results can relate to the two-layer model by Synofzik. The sentence about the limitation and utility of the questionnaire moved to revise the flow (original lines 115-116, revised lines 157-159)

We reduced the length of the introduction on Intentional Binding from the paragraph introducing the sense of agency research supporting the comparator view (originally lines 68-90, revised lines 68-84) and instead elaborated the section explaining why we did not include Intentional Binding in our study (original lines 167-179, revised lines 134-161). We believe that this change enhanced readability even with the redundant mentions of Intentional Binding. With this revision, our claim about originality (original lines 117-133) was divided into two (revised lines 120-133, 151-161). 

We believe these revisions address the comments effectively while enhancing the clarity and impact of our study’s contributions, and we sincerely hope this will be the final revision as we have worked diligently to incorporate the comments.

Response to the comments from Reviewer 2:

・Reviewer 2, Comment 1:

In lines 889-919 (clean copy), I think the authors should discuss oculomotor stability in relation to temporal attention. Although I do not deny the involvement of spatial attention, I think a discussion related to temporal attention, which is more relevant to the main hypothesis of the manuscript, is warranted. It has been noted that when a target appears at a certain time, various oculomotor events tend to stabilize as it approaches the expected time of target presentation, perhaps to get clearer image input at that time. For example, the pupil diameter, which plays a role in regulating the intensity of the image entering the eye, is stable at the time when a potential target is likely to appear (https://doi.org/10.1371/journal.pone.0276205). Also, changes in pupil diameter may lead to fine-scale variations in gaze position as measured by the eye tracker (https://doi.org/10.1371/journal.pone.0111197). It has also been reported that the gaze position becomes stable at the time when the target is about to appear (https://doi.org/10.1016/j.neuroimage.2018.09.026;
https://doi.org/10.1167/jov.23.14.1). I recommend primarily discussing these previously found phenomena in relation to the correlation between the display lag and fine-scale variation in gaze position.

・Response:

We appreciate the detailed suggestions, which we have carefully considered and integrated. In the revised manuscript, we have now included a comprehensive discussion on how temporal and spatial attention may relate to the delay’s impact on saccades under 1.5 degrees (lines 1000-1028 in the revised manuscript), while also reducing the emphasis on spatial attention to better align with the relevance of temporal attention in this context.

・Reviewer 2, Minor Comment 1:

Recently, an article has been published by the same author dealing with a similar topic (https://doi.org/10.3389/fpsyg.2024.1364076). It does not seem to deal with the exact same hypothesis, and there may not be any problem in terms of publication ethics. Nonetheless, it may be safer to cite the article. Also, please make sure that you are not using the same text or images.

・Response:

We agree that it would be good to clarify the difference from previous studies. We mentioned it briefly at the end of the introduction (lines 258-262 in revised)

・Reviewer 2, Minor Comment 2:

Because the analysis in the manuscript is voluminous, I suggest giving a brief summary of the specific hypotheses at the end of the introduction or at the beginni

---

## [Editor Report · Decision Letter 3]

23 Aug 2024

Sense of agency at a temporally-delayed gaze-contingent display

PONE-D-23-32262R3

Dear Dr. Kim,

We’re pleased to inform you that your manuscript has been judged scientifically suitable for publication and will be formally accepted for publication once it meets all outstanding technical requirements.

Kind regards,

Michael B. Steinborn, PhD

Section Editor

PLOS ONE
---

## [Editor Report · Acceptance letter]

29 Aug 2024

PONE-D-23-32262R3 

PLOS ONE

Dear Dr. Kim, 

I'm pleased to inform you that your manuscript has been deemed suitable for publication in PLOS ONE. Congratulations! Your manuscript is now being handed over to our production team.

Kind regards, 

on behalf of

Dr. Michael B. Steinborn 

Section Editor

PLOS ONE